# Extracting Probabilistic Knowledge from Large Language Models for Bayesian Network Parameterization

**Aliakbar Nafar**                                                             *nafarali@msu.edu*
*Michigan State University*

**Kristen Brent Venable**                                                      *bvenable@ihmc.org*
*Florida Institute for Human and Machine Cognition (IHMC)*
*University of West Florida*

**Zijun Cui**                                                                  *cuizijun@msu.edu*
*Michigan State University*

**Parisa Kordjamshidi**                                                        *kordjams@msu.edu*
*Michigan State University*

**Reviewed on OpenReview:** *https://openreview.net/forum?id=Fy3Byg3CVo*

## Abstract

In this work, we evaluate the potential of Large Language Models (LLMs) in building Bayesian Networks (BNs) by approximating domain expert priors. LLMs have demonstrated potential as factual knowledge bases; however, their capability to generate probabilistic knowledge about real-world events remains understudied. We explore utilizing the probabilistic knowledge inherent in LLMs to derive probability estimates for statements regarding events and their relationships within a BN. Using LLMs in this context allows for the parameterization of BNs, enabling probabilistic modeling within specific domains. Our experiments on eighty publicly available Bayesian Networks, from healthcare to finance, demonstrate that querying LLMs about the conditional probabilities of events provides meaningful results when compared to baselines, including random and uniform distributions, as well as approaches based on next-token generation probabilities. We explore how these LLM-derived distributions can serve as expert priors to refine distributions extracted from data, especially when data is scarce. Overall, this work introduces a promising strategy for automatically constructing Bayesian Networks by combining probabilistic knowledge extracted from LLMs with real-world data. Additionally, we establish the first comprehensive baseline for assessing LLM performance in extracting probabilistic knowledge.

## 1 Introduction

Bayesian Networks (BNs) are a powerful formalism for representing uncertainty and dependencies between events. The reliability of inference in BNs hinges on the accuracy of the conditional probability table (CPT) entries. CPTs are typically obtained by collecting data (Ji et al., 2015), which can be expensive or unattainable in domains where data is scarce (You et al., 2019; Longato et al., 2023). When data is limited, expert judgments are used as priors to be combined with the data for more accurate probability estimation (Mendes, 2014). However, experts are often unavailable (Das, 2004; Gao et al., 2019); when they are, their competence must be vetted (Hald et al., 2016), and their opinions must be aggregated (McAndrew et al., 2021) before their views can be used. Given these issues, we explore the capability of Large Language Models (LLMs) to act as experts for building BNs.

The potential of language models as sources for extracting factual knowledge has been demonstrated in several studies (Petroni et al., 2019; Roberts et al., 2020; AlKhamissi et al., 2022; Zhao et al., 2025). However, it

remains unclear whether LLMs possess the ability to generate meaningful *probabilistic estimates* for events and their relationships based on their internal knowledge. In this paper, we use the term *probabilistic estimation* to refer to assigning a specific probability to an uncertain proposition by LLM when utilizing its internal knowledge. For instance, consider the question, "What is the probability that a person who smokes cigarettes will develop cancer in their lifetime?" The answer to this question cannot be inferred from the question's context; however, a medical expert familiar with the literature might approximate 20%. Similarly, we expect an adept language model to produce a similar estimate. This contrasts with providing a *confidence score* for a concrete answer (Xiong et al., 2024) or solving a problem that has a known numerical solution.

We evaluate the probabilistic estimation capabilities of LLMs such as GPT-4o (OpenAI et al., 2024), Gemini 1.5 Pro (Gemini Team et al., 2024), Claude 3.5 Sonnet (Anthropic, 2024), and open-source model DeepSeek-V3 (DeepSeek-AI et al., 2024) and utilize their internal knowledge to construct domain-specific BNs with **discrete variables**. To provide a detailed analysis and clear evaluation of the parameter estimation, we assume the dependency structure within the BN is given. We note that the dependency structure extraction has been investigated in (Babakov et al., 2025) and the preliminary results are promising. In our setting, given the structure of a BN, LLMs are required to predict a probability distribution for each node, conditioned on its parent nodes. First, we analyze the quality of the initial distributions estimated by LLMs. Then, we investigate whether they can potentially function as expert-driven prior probabilities. We test this approach, which we denote as Expert-Driven Priors (EDP), by adjusting the LLM predictions with data samples, effectively applying a partial calibration to the model's initial estimates.

We use LLMs to estimate the CPTs of eighty real-world BNs collected from the literature. Using Kullback-Leibler (KL) divergence (Kullback & Leibler, 1951) as our metric, we show that EDP consistently improves LLM's predictions and offers a higher-quality prior than the conventional uniform baseline, which is employed when no extra information is available. Furthermore, our experiments indicate that even when the number of data samples is large, incorporating LLM priors improves performance. We further evaluate EDP on BNs used for classification, demonstrating that reducing KL divergence translates into higher downstream classification accuracy. These findings highlight the promise of leveraging LLMs as expert knowledge sources for probabilistic estimation across various real-world domains.

In summary, our contributions are as follows:

1) We introduce the first large-scale and comprehensive evaluation of LLM probabilistic estimation with real-world BNs. We investigate differences in LLM accuracy across domains and varying levels of network complexity, highlighting their effectiveness as probabilistic knowledge bases.

2) We show that LLM-predicted probabilities can serve as expert-driven priors, and combining them with data improves estimated probabilities compared to purely data-driven methods or using a uniform prior.

3) We evaluate our method on downstream classification tasks and demonstrate that the improved probability distributions lead to higher classification accuracy.

4) We introduce an automated procedure for parameterizing real-world BNs given the network structure[1].

## 2 Related Work

To the best of our knowledge, this is the first study to use LLMs to parameterize Bayesian Networks and to evaluate those parameters against ground-truth probabilities. Most prior work queries LLMs for a single number interpreted as a "confidence" in a class label, which is inherently different from probabilities (Levine, 2024). Among the few works that elicit probabilities, evaluation is only done at task accuracy because gold probabilities are unavailable. In contrast, we directly assess the quality of the learned CPTs via KL divergence to the original BN parameters, and in addition, examine downstream classification accuracy.

**Probability Estimation.** In the most relevant prior work, LLMs are prompted to produce probabilities of binary variables forming shallow BNs with a depth of only two, tailored toward classification tasks (Huang et al., 2025; Feng et al., 2025). Although they show improved downstream classification accuracy, they do

---

[1]Code and datasets are available at `https://github.com/HLR/llm-bn-parameterization`.

not evaluate the full probability distributions because gold probabilities are unavailable. Feng et al. (2025) only evaluates the correctness of the magnitudes of inferred-node probabilities, not having the ground-truth as a basis to evaluate the distributions. By contrast, we obtain a complete probability distribution for every discrete node, including those with more than two states, and evaluate it against real-world BNs of varying depth with known parameters. Further, we show how an improved estimation translates into better downstream classification tasks. Paruchuri et al. (2024) asks LLMs to calculate the probabilities for a range of values in a given distribution, but their dataset is limited to only 12 questions, and the queries are elementary with no conditional probabilities.

**Confidence Elicitation.** Confidence elicitation in LLMs has been studied in classification tasks, where a confidence score ranging from 0 to 1 is assigned to a discrete class label. Among these, Kadavath et al. (2022) treat the models as a white box and use their token probabilities to assess the confidence of a label. But, token likelihood indicates the model's uncertainty about the next token (Kuhn et al., 2023), rather than the confidence of the label. Consequently, Xiong et al. (2024); Yang et al. (2024) treat the model as a black-box and use its generated confidence to solve classification datasets. However, confidence is different from probability (Levine, 2024) and confidence elicitation does not directly apply to scenarios requiring a probability distribution across multiple states.

**Prior Elicitation.** Recent research has begun to use Generative AI for prior elicitation, primarily through data augmentation strategies. O'Hagan & Ročková (2025) introduces a non-parametric framework where AI-generated data serves as a base measure for Dirichlet process priors. In parametric settings, such as logistic regression, Gouk & Gao (2024) similarly prompts LLMs to generate synthetic datasets, which are then used to infer priors over regression coefficients. These methods largely rely on generating synthetic samples to calculate parameters. Our work differs by evaluating the LLM's ability to directly estimate the priors. By treating the model as a direct source of expert judgment rather than a synthetic data generator, our approach is significantly more cost-effective, as it retrieves parameters in a single query rather than incurring the token costs associated with iterative data generation. Related work also acquires priors by prompting for their parameters. Selby et al. (2025) elicit parametric priors such as Normal or Beta distributions for individual quantities and find that LLM-derived priors can sometimes help downstream analysis. Capstick et al. (2025) introduce AutoElicit, which obtains priors for a linear regression model parameters. In contrast, our work elicits full conditional distributions for BN nodes.

**Probabilistic Inference.** Probabilistic inference is closely related to our task and can be considered a natural extension of probabilistic estimation. Saeed et al. (2021); Nafar et al. (2024a) fine-tune BERT-based language models to perform probabilistic inference, while Nafar et al. (2024b) utilizes prompt engineering techniques to enable LLMs to conduct probabilistic inference. However, in all these approaches, the explicit probabilities are either provided in the text or learned from the dataset during fine-tuning without any estimation derived from the internal knowledge of language models.

**Zero-shot Regression.** Our method queries the LLMs for a numeric probability expressed in plain text, in a relatively similar setting to using LLMs for regression. Using LLMs for regression in a zero-shot setting is an emerging field, with a limited number of studies. Following Vacareanu et al. (2024), which shows that LLMs are capable regressors in a few-shot setting, Nafar et al. (2025) tests the regression capability of LLMs in a zero-shot setting (using internal knowledge) for realistic questions such as estimating the medical insurance cost based on age. However, they don't use any probability estimation. Requeima et al. (2024) moves closer to our setting by eliciting predictive distributions rather than point estimates, conditioned on numerical context and natural-language descriptions. Nevertheless, its goal remains probabilistic regression. In contrast, our work elicits conditional probability distributions for the rows of a Bayesian Network CPT under a known graph structure, and then incorporates these elicited distributions as structured priors for BN parameterization.

## 3 Problem Definition

The main problem addressed in this paper is parameterizing a Bayesian Network given its structure. We formally define the problem as follows, given the structure of a BN $\mathcal{G} = (\mathcal{V}, \mathcal{E})$ where $V$ is the set of nodes (random variables) and $E$ is the set of edges (dependencies among variables), the goal is to estimate the

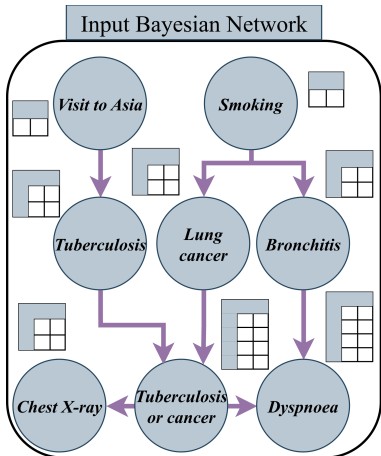

Figure 1: The Asia BN with its directed structure and associated CPTs. Given only the graph structure and the node descriptions, the goal is to estimate the entries of each CPT using LLMs.

parameters, that is, Conditional Probability Tables $\mathcal{G}_\theta$ of this network with the help of LLMs. Figure 1 illustrates this task with the Asia BN (Lauritzen & Spiegelhalter, 1988), where each node has an associated CPT whose entries must be estimated. After assigning the parameters, we compare the resulting distribution to ground-truth values of the original BNs.

Once a BN is parameterized, it can also serve as a classifier by designating one node as the target variable and using probabilistic inference to predict its value given evidence on the remaining nodes. For instance, in the Asia network shown in Figure 1, *Visit to Asia* is the target variable and predicts whether a patient has recently visited Asia based on observed symptoms such as dyspnoea and chest X-ray results. It is worth noting that not all BNs are suited for classification; this use case applies primarily to networks where a meaningful target variable can be identified, such as BNs with a Naive Bayes structure.

## 4  Methodology

As illustrated in Figure 2, our BN parameterization pipeline has two steps. First, we query an LLM to use its internal probabilistic knowledge to generate probability estimates. Next, we use these estimates in our **Expert-Driven Priors (EDP)** framework, which treats the LLM outputs as priors and refines them with data to produce the final parameters.

### 4.1  Extracting Probabilistic Knowledge

In the first stage of our BN parameterization pipeline, we use LLMs to acquire probabilistic estimates for every row of the CPTs. Because many nodes are multi-state, asking for a single probability score for each node is impractical. So we use two prompting schemes: **1) FullDist**, where the entire distribution is obtained from the LLM at once in the form of a tuple, e.g., $(0.70, 0.20, 0.10)$. **2) SepState**, where each state is queried independently. Figure 2 depicts the SepState scheme, where the process starts with a prompt template that describes the node and its parents in a natural language format. The descriptions of these nodes, combined with the LLM instructions, are appended to each question presented to the LLM. Each question explicitly defines the states of the node of interest as well as the states of its parent nodes, and poses a probabilistic query based on these assigned states. The LLM is also instructed to articulate its reasoning before generating a probability value. The final answer is extracted as a numerical probability from the output text. Since the raw numeric outputs may not sum to one, they are normalized to form a valid distribution over the node's states. For a node with $m$ states, the model might produce values $p_1, p_2, \ldots, p_m$ that sum to $S = \sum_{i=1}^{m} p_i$. To convert these values into a valid probability distribution, we divide each one by $S$. This normalization

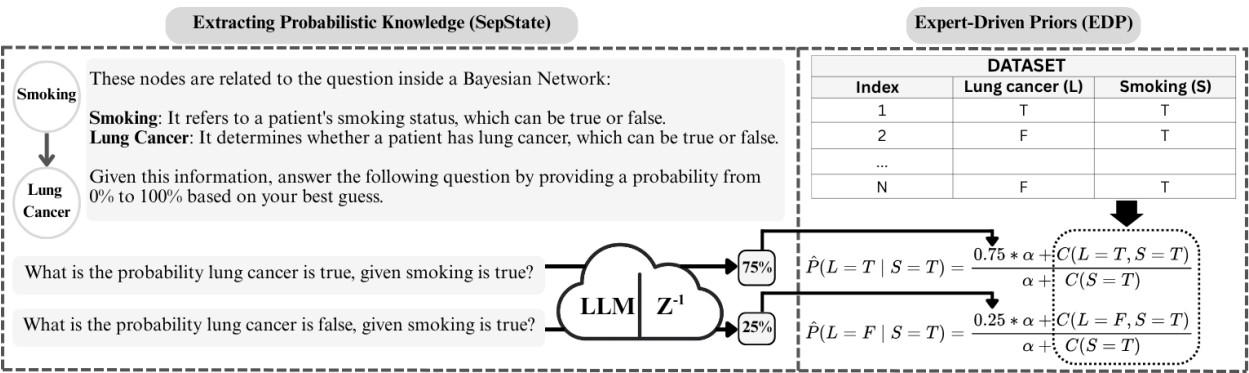

Figure 2: **Two-stage parameterization pipeline.** *SepState (Left Panel):* For each parent configuration, the LLM is prompted with natural-language descriptions of the node and its parents and queried once per state. The answers are subsequently normalized ($z^{-1}$) into a valid conditional distribution, e.g., (75%, 25%). *EDP (Right Panel):* The LLM-derived prior distribution is translated into pseudocounts and fused with empirical counts ($C$) to give the posterior estimates. This treats the LLM as a probabilistic expert whose influence is controlled by the hyperparameter $\alpha$.

step can be interpreted as taking the ratio of each state's assigned likelihood relative to the sum of all states, effectively **preserving the proportions** while enforcing a valid distribution.

FullDist follows the same prompting template, but the LLM is instructed to return the entire distribution in one shot. For example, instead of the two questions in Figure 2, the LLM is asked "What is the probability distribution of lung cancer given that smoking is true?" Thus, the LLM returns a tuple $(p_1, \dots, p_m)$ in a single response. Unlike SepState, the FullDist scheme is concise and requires only one query, independent of the number of the node's states. However, the autoregressive decoder conditions each number on the previous ones, so early outputs can bias later entries. Concentrating all states in a single prompt could also be too complicated for the LLM. In our experiments, we compare SepState and FullDist schemes empirically.

As noted above, our prompts include brief definitions of each node and its state space. However, this information can sometimes be extracted from the questions themselves. For example, the meanings of "Lung Cancer" and "Smoking" are commonsense knowledge and known to the LLMs. Also, the LLM can infer that their values are binary, based on the given true/false assignments. However, in cases where the semantic meaning or value sets of nodes are not immediately clear, explicit descriptions are needed. For instance, a node named "X1" must have a clearly stated meaning, such as: *"Represents a lack of supervision and policy guidance, which may lead to the use of unqualified oil. This node can take True or False values."* Similarly, while the semantic meaning of the "Construction Year" node is self-descriptive, its possible states are ambiguous and need an explanation such as: *"This node indicates the time period in which the building was constructed, with possible values being 1930-1955, 1955-1960, 1960-1968, 1968-1975, and 1975-1980."* We later test the usage of the contextual descriptions of node meanings and state sets by LLMs in an ablation study to measure their effect on the LLM's probability predictions.

## 4.2 Expert-Driven Priors

Expert opinion and large datasets are either costly or difficult to obtain in many practical scenarios. When only limited data is available, incorporating expert prior knowledge can particularly help offset the shortage of empirical data, thereby improving the estimated probability distributions. We propose that LLMs can approximate these prior distributions based on their knowledge. In our approach, Expert-Driven Priors (EDP), we combine the LLM-derived probabilities with the empirical distribution estimated from data by using the estimated priors as pseudocounts (Zhai & Lafferty, 2001).

For a node in the BN with $m$ discrete states, conditioned on a specific parent configuration, the LLM provides a normalized prior distribution $q_1, \dots, q_m$ as shown in the left panel of Figure 2. Then, we use these priors and incorporate data to calculate the final probability distribution. Let $c_1, \dots, c_m$ denote the

observed counts listed in the data table on the right side of Figure 2. Each prior probability $q_i$ is converted into $\alpha q_i$ virtual observations, where $\alpha \in [0, \infty)$ is a hyperparameter. Larger $\alpha$ values place more weight on the LLM prior, whereas smaller values defer to the data. Adding these pseudocounts to the empirical counts gives the posterior estimate displayed inside the dashed box of Figure 2:

$$p_i = \frac{\alpha q_i + c_i}{\alpha + \sum_{j=1}^{m} c_j}, \qquad i = 1, \ldots, m.$$

The numerator combines prior belief ($\alpha q_i$) with empirical evidence ($c_i$), while the denominator $\alpha + \sum_j c_j$ normalizes the resulting probabilities. In this manner, we incorporate LLM predictions in the same principled way one would incorporate probabilities elicited from a human expert.

## 5 Experiments

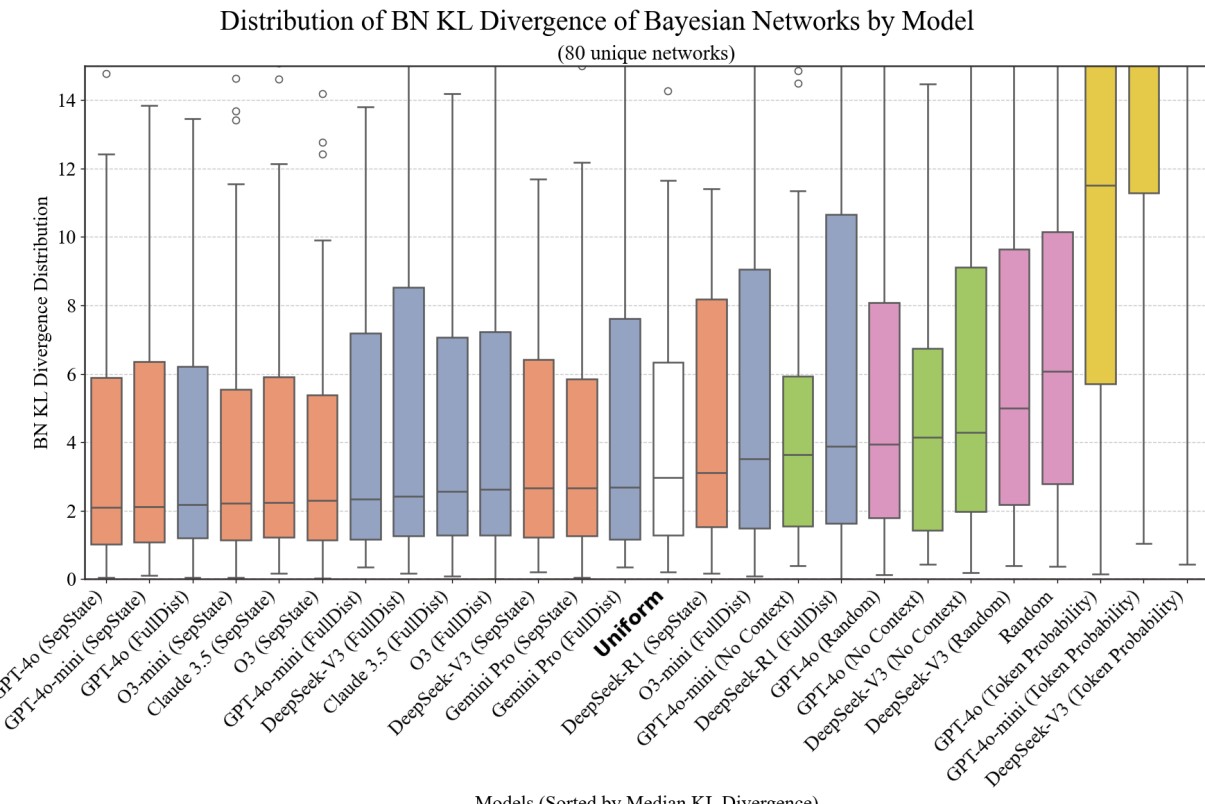

Figure 3: Boxplot showing distribution of BN KL divergence values across eighty unique BNs for various models, sorted by their median KL divergence. Lower values indicate better alignment with ground-truth CPTs.

### 5.1 Dataset of Eighty Bayesian Networks

Our experiments use bnRep (Leonelli, 2025), an open-source collection designed to facilitate research, teaching, and practical applications related to BNs, addressing the significant shortage of comprehensive BN repositories. Implemented as an R package, bnRep includes over 200 well-documented Bayesian Networks sourced from more than 150 academic publications, mainly recent studies published from 2020 onwards. Each BN entry has accompanying characteristics extracted from the original publications about the publication and the BN such as domain of the paper and the structure and size of the BN. Refer to Appendix A for the full list of characteristics and their descriptions.

<table>
<tr><td>

**Prompt Template for the SepState Scheme**

These nodes are related to the question inside a Bayesian Network:

(List of nodes and their descriptions)

Given this information, answer the following question by providing a probability from 0 to 1 based on your best guess (you need to make a lot of estimations since the given information is limited). Your answer should include your reasoning and, at the end, a sentence that says 'The probability of the question is: ' followed by the probability.

(Probabilistic query in the form of a question)
(E.g., given that ... what is the probability that X is x? )

</td><td>

**Prompt Template for the FullDist Scheme**

These nodes are related to the question inside a Bayesian Network:

(List of nodes and their descriptions)

Given this information, answer the following question by providing the probability distribution of the node. Your answer should include your reasoning and, at the end, a sentence that says 'The probability distribution of the node is: ' followed by the probabilities given in a tuple with each probability representing a state in the given order.

Order of states: (List of the states of the node)

(Query to request the full distribution)
(E.g., given that ... what is the probability distribution of X? )

</td></tr>
</table>

Figure 4: Prompt templates for the SepState and FullDist schemes. Both templates supply the LLM with node descriptions and specific formatting constraints, but they differ in their target output: the SepState scheme (left) instructs the model to estimate a single probability value (0 to 1), whereas the FullDist scheme (right) elicits the complete probability distribution tuple for all the possible states of the target node.

To utilize the bnRep dataset, we first converted the BNs from the R package into Python-compatible format using the pgmpy library (Ankan & Panda, 2015). Next, we filtered the networks, selecting only those containing discrete CPT values, as these comprise the majority of the BNs. We excluded networks with more than 50 nodes due to practical and cost considerations, noting that only a few exceed this threshold. The most substantial reduction, however, came from removing BNs with incomplete CPT information. After applying all these criteria, we arrived at a final dataset of eighty BNs, which remains sufficiently large for our evaluation purposes. To acquire the description of nodes for each BN, we first retrieved the PDFs of the referenced documents that detail each BN. We then implemented a Python script leveraging GPT-4o, which, given a PDF and the extracted nodes and states of the BN (obtained through the pgmpy library), automatically generated a Python dictionary describing each node and its associated states. The majority of generated dictionaries were accurate and required minimal modifications, while a few necessitated manual adjustments to ensure correct formatting and accuracy.

## 5.2 Metrics and Baselines

For LLMs, we use GPT-4o and its mini variant (OpenAI et al., 2024), along with Claude 3.5 Sonnet (Anthropic, 2024), Gemini 1.5 Pro (Gemini Team et al., 2024), and DeepSeek-V3 (DeepSeek-AI et al., 2024). We include reasoning models, o3 and its mini variant (OpenAI, 2025), and DeepSeek-R1 (DeepSeek-AI et al., 2025). The prompt templates used for testing the LLMs are presented in Figure 4. These templates provide the LLMs with the relevant node descriptions, followed by strict formatting instructions to extract the probability parameters. In instances where the response violates formatting constraints, such as yielding a negative or out-of-range value, the LLM is repeatedly prompted until a valid probability number is successfully extracted[2].

We evaluate the LLM's estimated BN parameters using Kullback-Leibler (KL) divergence (Kullback & Leibler, 1951) compared to ground-truth parameters in bnRep. Specifically, we report the *BN KL divergence*, defined as the KL divergence computed over the BN variables' joint distribution, evaluating the resulting BN's overall quality. Refer to Appendix B.1 for an overview of KL divergence and BN KL divergence, and to Appendix C for LLM and hyperparameter configurations, specifically detailing our rationale for selecting $\alpha$ and its impact on performance.

We evaluate our methods against multiple baselines. All outputs from these baselines are normalized as necessary to ensure valid probability distributions. These baselines are: (1) **Random** number generator; (2) **Uniform** baseline generating equal probabilities for each row of the CPT, providing basic, uninformed

---

[2]In practice, formatting violations were rare. Re-prompting was primarily necessitated by system issues such as API timeouts.

estimations; (3) **LLM (Random)** baseline involving intentionally incorrect queries, where the original variable names are randomly replaced. This is done to assess whether LLMs utilize the content of the provided questions to generate their answers. For instance, we query the LLM "What is the probability that construction time is true given that lung cancer is true?" instead of the correct question regarding smoking; (4) **LLM (No Context)** baseline in which queries are presented without contextual explanations, exploring scenarios where the node meanings and number of states cannot be directly inferred from the in-context information; (5) **LLM (Token Probability)** baseline, which directly uses the LLM's probabilities assigned to tokens representing node states (e.g., probability of generating the token "True"), rather than explicitly generated numerical probabilities extracted from the model's textual responses; (6) **MLE-#** is a statistical baseline obtained by maximum likelihood estimation using # data samples; (7) **Uniform-#** applies the same pseudocount updating method as EDP with # data samples, but uses a uniform prior instead of LLM-derived probabilities. Data samples are obtained from the ground-truth BN where the BN is sampled # times using forward sampling. For results obtained using other sampling methods, refer to Appendix D.

### 5.3 Can LLMs Estimate Probabilities Using Their Internal Knowledge?

To evaluate how SepState and FullDist compare to other baseline models, we analyze the distributions of *BN KL divergence* across all eighty BNs, as depicted in Figure 3. The worst-performing models, which perform worse than random, are the "Token Probability" models, shown with yellow boxes. This aligns with previous research, which found that raw token probabilities from LLMs alone are insufficient for effective uncertainty/probability estimation (Xiong et al., 2024) and require additional processing steps like fine-tuning (Tao et al., 2024). The next weakest results are observed among the random generators, alongside the baselines that do not receive the context of nodes and their states. In our experiments, LLM (Random) models slightly surpass the outputs of the Random number generator baseline. However, these improvements only reflect the non-uniformity of random number generation by LLMs, influenced by factors such as text-generation sampling methods and model architecture choices (Hopkins et al., 2023). Of all the baseline models, the uniform predictor performs best. This result aligns with information theory, which suggests that, in the absence of knowledge, a uniform distribution naturally provides the lowest KL divergence based on uncertainty (Cover & Thomas, 2006).

Both FullDist and SepState outperform the uniform baseline in all non-reasoning models. o3 and DeepSeek-R1 trail their non-reasoning counterparts, hinting that high levels of reasoning do not translate to better probabilistic estimation. While the FullDist yields informed estimates, it consistently falls short of SepState with a higher median KL divergence and a greater standard deviation, except in the DeepSeek model. These results confirm the shortcomings of the FullDist scheme and establish SepState as a superior method to extract a full probability distribution. Overall, these results demonstrate the capability of LLMs to provide meaningful probability distributions, laying the groundwork for treating them as informative priors in EDP. A one-sided paired Wilcoxon signed-rank test (Wilcoxon, 1945), comparing each model's per-BN KL divergence against the uniform baseline, confirms statistical significance at $p < 0.01$ for GPT-4o (SepState and FullDist), o3-mini (SepState), o3 (SepState), Claude 3.5 Sonnet (SepState), and GPT-4o-mini (SepState). Gemini 1.5 Pro (SepState and FullDist) achieves $p < 0.05$, while the remaining configurations that outperform uniform, namely GPT-4o-mini (FullDist) and DeepSeek-V3 (SepState and FullDist), achieve $p < 0.1$.

### 5.4 Can LLMs' Probability Distribution Estimates Serve as Expert Priors?

In this section, we evaluate the effectiveness of EDP, which combines the LLM-derived distributions with empirical data, using priors as pseudocounts. Figure 5 displays the distributions of BN KL divergences obtained by combining various sample sizes of data with GPT-4o priors (EDP-#) and Uniform priors (Uniform-#), where # determines the number of sampled data. The Uniform prior is the conventional choice in the absence of prior information and serves as a baseline prior in our experiments.

EDP predictions consistently outperform the Uniform-# baseline, proving its use as a better prior. The advantage of EDP is most notable at smaller sample sizes, i.e., 3 to 30 samples. The combination of even minimal data in EDP significantly outperforms SepState and the MLE model with 30 data samples. Additionally, EDP still improves the median KL divergence when more data is available, e.g., 1k samples.

It also effectively reduces the standard deviation of the predictions, enhancing model robustness. This improvement in median KL divergence at large sample sizes occurs because nodes with unlikely parent combinations rarely receive data. When the dataset is sufficiently large, such as 10k samples in our setting, MLE achieves the best KL divergence, as expected. However, this outcome relies on our forward sampling procedure, which ensures every node of the BN is sampled. In the real world, data is often sparse or biased. In this case, even with 10k samples, EDP can still provide benefits, as we will show in the next section.

For brevity, the main text reports EDP results only for GPT-4o. The corresponding EDP plots for every other LLM prior that outperforms the Uniform baseline in Figure 3 display the exact same pattern as GPT-4o and are included in Appendix D. These findings confirm that using LLM predictions as expert-driven priors improves the performance and robustness of parameter estimation in Bayesian Networks.

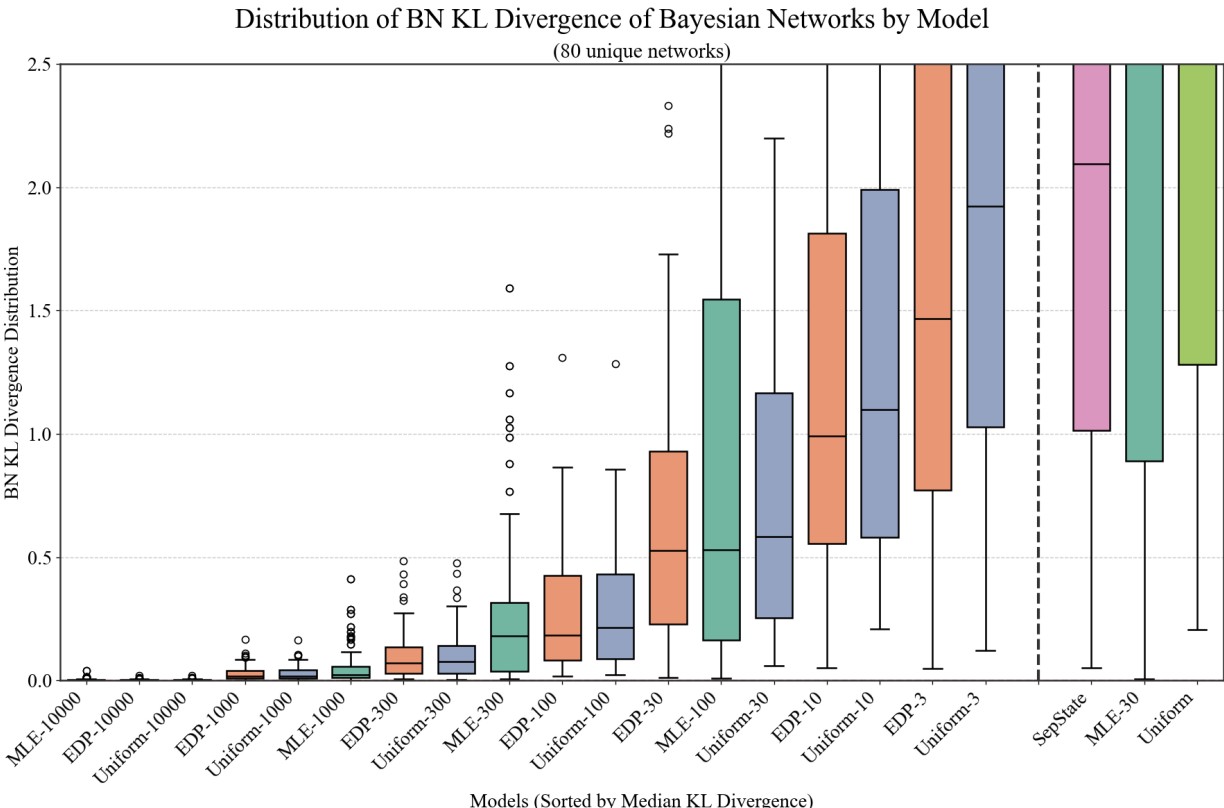

Figure 5: Boxplot of the distribution of BN KL divergence over eighty networks, contrasting models with GPT-4o priors (EDP-#), uniform priors (Uniform-#), and data-only probability estimates (MLE-#). The numeral after the "-" denotes the sample size that is used with the method before the "-".

## 5.5 What Is EDP's Impact on Downstream Tasks?

We have shown that EDP lowers BN KL divergence, but this alone does not guarantee better performance on downstream tasks. To test whether improved probability estimates translate into downstream gains, we focus on classification, one of the common BN applications. However, the majority of BNs in the bnRep are not intended for classification tasks. A BN designed for classification typically features a central target node, where the primary objective is label prediction. Networks with a Naive Bayes structure are a prime example of this type of BNs. Following the experimental setting in (Carli et al., 2022), we select nine real-world datasets that are routinely used for BN-based classification studies, each with one target random variable. Every dataset is stratified into 80% training and 20% test splits. For each dataset, two BN structures are used: (i) a structure discovered via Hill Climbing, learned based on the full training data,

| Dataset | Baseline | | | Hill Climbing | | | | | | Naive Bayes | | | | | | |
| | Rand | CoT | SS | Full Data | | 20 Samples | | 10 Samples | | SS | Full Data | | 20 Samples | | 10 Samples | |
| | | | | MLE | EDP | MLE | EDP | MLE | EDP | | MLE | EDP | MLE | EDP | MLE | EDP |
|---|---|---|---|---|---|---|---|---|---|---|---|---|---|---|---|---|
| HV84* | 0.52 | 0.50 | 0.24 | **0.94** | 0.91 | **0.87** | 0.80 | 0.78 | **0.85** | 0.40 | **0.94** | 0.91 | **0.92** | 0.90 | **0.91** | 0.87 |
| PhDA* | 0.38 | 0.39 | 0.41 | 0.38 | **0.41** | 0.35 | **0.41** | 0.34 | **0.41** | 0.45 | **0.44** | **0.44** | 0.36 | **0.41** | 0.34 | **0.41** |
| Pokemon | 0.50 | 0.43 | 0.23 | **0.62** | **0.62** | **0.62** | **0.62** | **0.54** | **0.54** | 0.48 | **0.62** | 0.60 | 0.59 | **0.60** | 0.53 | **0.54** |
| Titanic | 0.51 | 0.71 | 0.42 | **0.42** | **0.42** | **0.42** | **0.42** | **0.42** | **0.42** | 0.57 | 0.11 | **0.55** | 0.22 | **0.57** | 0.42 | **0.56** |
| CAD1 | 0.54 | 0.55 | 0.87 | **0.83** | **0.83** | 0.74 | **0.85** | 0.63 | **0.84** | 0.77 | 0.81 | **0.85** | 0.83 | **0.86** | 0.80 | **0.84** |
| CAD2 | 0.44 | 0.44 | 0.76 | 0.65 | **0.76** | **0.76** | **0.76** | 0.60 | **0.76** | 0.50 | **0.86** | 0.79 | 0.73 | **0.78** | 0.77 | **0.78** |
| Covid | 0.54 | 0.64 | 0.74 | 0.71 | **0.73** | 0.70 | **0.72** | **0.72** | **0.72** | 0.72 | 0.71 | **0.72** | 0.71 | **0.72** | 0.69 | **0.72** |
| Puffin | 0.50 | 0.43 | 0.63 | **1.00** | 0.93 | **0.99** | 0.91 | **0.97** | 0.91 | 0.78 | **0.93** | 0.85 | **0.90** | 0.88 | 0.87 | **0.88** |
| Traject* | 0.59 | 0.78 | 0.87 | **0.87** | **0.87** | 0.75 | **0.86** | 0.68 | **0.86** | 0.80 | **0.87** | **0.87** | **0.86** | **0.86** | 0.85 | **0.86** |
| Average | 0.50 | 0.54 | 0.57 | 0.71 | **0.72** | 0.69 | **0.71** | 0.63 | **0.70** | 0.61 | 0.70 | **0.73** | 0.68 | **0.73** | 0.69 | **0.72** |

Table 1: Macro $F_1$ classification scores on nine datasets for BN classifiers. The baselines include Random (Rand) and Chain-of-Thought (CoT) methods. The rest of the columns include three methods, *SepState (SS)* using GPT-4o, *MLE*, and *EDP*, under two graph structures, Hill Climbing and Naive Bayes. Full Data uses the entire training split, whereas 20 and 10 Samples simulate low-resource regimes. Within each data regime, the higher of *MLE* vs. *EDP* is bolded. * datasets HV84, PhDA and Traject refer to HouseVotes84, PhDArticles and Trajectories, respectively.

and (ii) a simpler Naive Bayes structure. Besides the full-data regime, we simulate low-resource scenarios by using the same network structure but restricting the training set to 20 and 10 samples for parameter estimation, and averaging the results over 5 random runs. We evaluate three LLMs, GPT-4o, GPT-4o-mini and DeepSeek-V3, using Chain-of-Thought (CoT) (Wei et al., 2022), SepState, MLE, and EDP.

Table 1 reports Macro $F_1$ scores of GPT-4o over nine datasets and compares them for each network structure. EDP improves over MLE, especially when data is scarce. The advantage of EDP is most evident in the derived structure from Hill Climbing because learning parameters for its complex structure is more difficult with scarce data. With full training sets, EDP still edges out or ties MLE in 12 out of 18 cases. The trend holds when switching from a Hill Climbing structure to Naive Bayes, and EDP surpasses MLE at every data size. This trend also remains consistent when we test GPT-4o-mini and DeepSeek-V3 (tables are moved to Appendix D for space). The only difference is that we get better classification results with EDP using the FullDist scheme for DeepSeek-V3, which achieved a better parameter estimation based on the obtained KL divergence as shown in Figure 3.

The improvement provided by EDP in low-data regimes is statistically significant ($p < 0.05$). However, in the full-data regime, although EDP generally improves results, the difference is not statistically significant. Detailed calculations using the Wilcoxon signed-rank test (Wilcoxon, 1945) are provided in Appendix D. This observation mirrors the trend in Figure 5, which suggests that priors may introduce bias and degrade performance once sufficient data is available. In our nine datasets, some already achieve very high performance without EDP. For example, in datasets like HouseVotes84 and Puffin, the MLE baseline already nears the optimal ceiling (0.94–1.00). Crucially, however, EDP proves robust even in these cases, avoiding any significant drop in performance. Taken together, these findings corroborate the central claim that LLM-derived probability estimates are not only closer to ground truth compared to other alternative baselines but are also effective in downstream tasks.

## 6 Discussion

**Quality of Individual Distributions Versus the Entire Bayesian Network.** In our experiments, we used the BN KL divergence metric to evaluate the quality of the predicted BN parameters. However, this metric assesses the entire BN, meaning that a few poorly predicted nodes might disproportionately influence the evaluation. To address this limitation, we also analyzed the *CPT KL Divergence*, which computes the

average KL divergence across all individual CPT rows within each BN. This alternative measure evaluates the quality of individual distributions rather than the BN as a whole. Using CPT KL divergence, we observed that the overall trends of our results remained consistent. Additional diagrams illustrating these findings are provided in Appendix D.

**Performance Variations Among LLMs Across Different Bayesian Network Domains.** Among the evaluated LLMs, GPT-4o consistently exceeds the performance of other LLMs, though specific models perform better within specialized domains. For instance, Claude 3.5 Sonnet, Gemini 1.5 Pro, and GPT-4o achieve the best results on BNs related to engineering, business, and medical domains, respectively. Furthermore, there are inherent differences in prediction behavior among these LLMs, likely attributed to their respective training methodologies. Specifically, Claude 3.5 Sonnet performed best on BNs with low entropy probabilities, but showed the poorest performance on BNs with high entropy probabilities among the LLMs, indicating an overly confident prediction behavior. In contrast, Gemini 1.5 Pro showed the opposite trend, whereas GPT-4o had a more balanced prediction profile. For a detailed breakdown of BN KL divergence across all 13 domain areas in our BNs, refer to Appendix F.

**Handling Larger Parent Sets and States.** Intuitively, it is expected that LLMs may struggle to provide informed predictions for more complex queries involving nodes with many parent nodes or states. We use the CPT KL divergence metric to assess predictions among these nodes, which averages the KL divergence across all individual CPT rows rather than evaluating the entire BN. In our experiments with realistic BNs, LLM performance in these scenarios still surpassed our baseline models. LLMs consistently outperformed baselines in queries involving up to 5 parent nodes. Additionally, the LLMs performed better than baselines for nodes with 2 or 3 states. Nodes exceeding 3 states are rare in realistic BNs. Only 4 BNs had nodes restricted to 4 states, whereas 11 featured nodes with 5 states. Within these BNs, except for the "DustExplosion" BN, the LLMs consistently outperformed baseline methods. "DustExplosion" contained nodes with both 4 and 5 states and is designed to predict explosion probabilities in industrial environments, which proved challenging for the LLMs. For detailed results demonstrating the performance with varying numbers of parent nodes and states, refer to Appendix E.

**Effect of Prompt Instructions on Probabilities.** Since LLMs are known to be sensitive to prompt phrasing, we investigated the robustness of our probability estimates to variations in the prompt template. Specifically, we tested several alternative configurations, including instructing the LLM to provide only the final numerical answer without any reasoning, as well as augmenting the prompt with additional context beyond the node-level descriptions, such as a high-level overview of the BN's structure and purpose. None of these variations affected the aggregate results across the eighty BNs. This robustness is consistent with related work on LLM-based regression, which similarly concludes that chain-of-thought reasoning does not significantly alter the predicted numerical output when aggregated over a dataset (Nafar et al., 2025).

**Cost Considerations for LLM-Based BN Parameterization.** Given the significant costs associated with LLMs, it is important to consider methods for reducing their inference cost. Although SepState consistently outperforms FullDist in 7 out of 8 models, achieving lower median KL divergence and improved stability, FullDist requires only a single query per CPT row, independent of the state count, and still outperforms the uniform baseline in most models. We therefore recommend SepState when accuracy is paramount, while FullDist remains a viable cheaper alternative. Furthermore, as noted earlier, suppressing chain-of-thought reasoning yields comparable results while substantially reducing inference cost.

Even with these cost reduction methods, utilizing frontier LLMs to parameterize large BNs remains a substantial expense. However, when evaluated against traditional alternatives for BN parameterization, LLMs frequently emerge as the more cost-effective and viable solution. Conventional methods typically rely on large-scale, domain-specific data collection, which is cost-prohibitive and often infeasible at scale, particularly for rare diseases or novel domains, or elicitation from human domain experts, who are costly, scarce, and require both competence vetting and opinion aggregation (Hald et al., 2016; McAndrew et al., 2021). Ultimately, the value of improved priors is application-dependent. In settings where domain data is genuinely unavailable or where small accuracy gains have substantial downstream impact, even modest improvements over the uniform baseline can readily justify the additional financial or computational expense of using LLMs.

**EDP's Application for Automated Bayesian Network Construction.** The demonstrated capability of LLMs, particularly GPT-4o, in the EDP method has significant implications for automating BN construction. Traditionally, parameterization of BNs relies heavily on expert input, making the process labor-intensive, costly, and dependent on the availability of reliable experts. Utilizing an LLM proficient across diverse domains removes these barriers and holds potential for automation in BN parameterization.

**EDP with Small Data for Extraction of Probabilistic Knowledge.** We showed the potential of using LLM predictions in place of expert-driven priors for constructing BNs. As shown in Figure 5, EDP with just a few data samples, such as 3, yields a lower BN KL divergence than both MLE trained on more data, like MLE-30, or the LLM-only (SepState). These results highlight a promising application wherein LLMs use minimal external data points to rapidly refine their predictions for probabilistic queries. Such a small amount of data could be supplied to the LLM in various ways, such as being obtained online by querying information from publicly available sources (e.g., occurrences of lung cancer among smokers). EDP not only improves data efficiency and outperforms uniform baseline methods but also provides an exciting possibility for real-time improvement of LLM-generated probabilistic estimates with minimal data input.

## 7    Conclusion and Future Work

In this work, we demonstrate that modern LLMs can effectively produce conditional probabilities to be used as expert-driven priors for Bayesian Network parameterization. EDP proved superior to the approach of using Uniform priors and purely data-driven approaches, especially in a low-data regime. Furthermore, we show that employing EDP to estimate parameters of the BNs improved the accuracy in downstream tasks. In conclusion, this study introduces a novel pipeline for parameterizing BNs using LLMs. Using LLMs as a resource compensates for the lack of data and reduces the need for costly domain experts. We also establish the first comprehensive framework for evaluating the probabilistic knowledge of LLMs with real-world probabilistic Bayesian Networks.

Future work will focus on advancing toward a fully automated framework for BN construction. In this regard, the key challenge lies in automating the structure learning component. Although preliminary efforts have been made in this area, there is significant potential to create an end-to-end pipeline using LLMs that generates a BN structure, parameterizes it, and systematically evaluates its performance.

## Limitations and Ethical Considerations

Despite the promising results of our methodology, the effectiveness of LLM-derived priors remains inherently domain-dependent. As observed in our BNs, GPT-4o demonstrated superior performance in medical contexts, whereas Gemini 1.5 Pro proved more adept at business applications. This variability dictates that any selected LLM must be vetted for its target domain prior to deployment, much like evaluating a human expert's credentials (Hald et al., 2016). Moreover, while our results show that LLM-derived priors consistently outperform uninformed baselines, merely surpassing a uniform distribution does not inherently qualify these models for safety-critical applications like healthcare or industrial engineering. In such high-stakes settings, elicited priors should be treated strictly as strong initial estimates rather than definitive conclusions.

Beyond domain suitability, the underlying training data of these models presents two distinct challenges. First, because the bnRep dataset is publicly available, we cannot definitively rule out data contamination within the LLMs' pre-training corpora. That said, the process of eliciting LLM priors and fusing them with observational data presents substantial challenges that must be solved regardless of whether the models have seen the evaluation data. Second, LLMs are known to encode the social and demographic biases present in their training data. If left unaddressed, these biases can seamlessly propagate through the elicited priors into the resulting BN's decision-making. Consequently, practitioners must audit these biases before using the priors if possible.

## Acknowledgments

This project is partially supported by the Office of Naval Research (ONR) grant N00014-23-1-2417. Any opinions, findings, and conclusions or recommendations expressed in this material are those of the authors and do not necessarily reflect the views of Office of Naval Research. We thank the anonymous reviewers for their valuable feedback and suggestions, which have helped improve this work.

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

# A Dataset Pre-processing

## A.1 bnRep Dataset Overview

The bnRep dataset (Leonelli, 2025) is an open-source R package providing a curated repository of over 200 Bayesian Networks (BNs) drawn from more than 150 academic publications, predominantly from 2020 onwards. Each BN entry is annotated with a set of characteristics extracted from its source publication, which we list and describe below:

- **Name**: A short identifier for the Bayesian Network.

- **Type**: The network's type of random variables (discrete, continuous, mixture).

- **Structure**: Indicates how the network's structure was obtained.

    - *Knowledge*: The structure is built from well-established domain knowledge.

- – *Data*: The structure was learned from a dataset.
- – *Fixed*: A predefined structure that is neither purely elicited from experts nor learned from data (often a canonical or standard network).
- – *Synthetic*: The structure was generated artificially (e.g., for algorithm testing).
- – *Expert*: The structure is directly elicited from domain experts.
- – *Mixed*: The structure is derived through a combination of sources.

- **Probabilities**: Indicates how the CPTs were obtained:

  - – *Data*: Parameters estimated from empirical data.
  - – *Knowledge*: Parameters derived from well-established theoretical or domain-specific information.
  - – *Mixed*: A combination of data-based estimation and expert input.
  - – *Synthetic*: Artificially generated parameters for testing or demonstration.
  - – *Expert*: Parameters directly elicited from domain experts.

- **Graph**: Describes any special structural characteristic of the network graph. For example:

  - – *Generic*: No particular restriction or canonical form.
  - – *Naive Bayes*: A star-shaped structure often used for classification tasks.
  - – *Reverse Naive Bayes*: The class label is modeled as a child of all other variables, reversing the direction of edges in the standard Naive Bayes structure.
  - – *K-Dep*: Each feature depends on the class and up to K other features.
  - – *Tree*: A graph with each node having exactly one parent (except the root).
  - – *Reverse Tree*: A tree with reversed edges, placing the class node at the leaves.
  - – *TAN*: an extension of Naive Bayes that allows each variable to have one additional parent, forming a tree among the predictors for greater flexibility.

- **Area**: The domain of the BN (e.g., Medicine, Engineering, Environmental Science).

- **Nodes**: The total number of random variables (nodes) in the Bayesian Network.

- **Arcs**: The total number of directed edges (arcs) in the network.

- **Parameters**: The total number of probability entries in the CPTs.

- **Avg. Parents**: The average number of parent nodes per variable.

- **Max Parents**: The maximum number of parents any single node has in the network.

- **Avg. Levels**: The average number of discrete states (levels) per node.

- **Max Levels**: The maximum number of states among all nodes in the network.

- **Average Markov Blanket**: The average size of the Markov blanket for each node, which consists of the node's parents, children, and the children's other parents.

- **Year**: The year of publication associated with the BN's reference.

- **Journal**: The venue where the Bayesian Network was published or described.

- **Reference**: The bibliographic reference describing the BN in detail.

# B  KL Divergence

## B.1  Kullback-Leibler (KL) Divergence Overview

Kullback-Leibler (KL) divergence (Kullback & Leibler, 1951) measures how one probability distribution $p$ diverges from a reference distribution $q$. For a discrete random variable $X$,

$$D_{\mathrm{KL}}\big(p(X) \,\|\, q(X)\big) = \sum_{x \in \mathcal{X}} p(x) \, \log \frac{p(x)}{q(x)}.$$

For continuous variables, the sum is replaced by an integral. KL divergence satisfies $D_{\mathrm{KL}}(p\|q) \geq 0$ and equals zero iff $p = q$. It is asymmetric ($D_{\mathrm{KL}}(p\|q) \neq D_{\mathrm{KL}}(q\|p)$ in general). Intuitively, it quantifies the expected extra amount of information (in nats or bits) required to encode samples from $p$ using a code that is optimal for $q$. In our implementation, all computations use base-2 logarithms (bits). To avoid undefined terms when $q(x) = 0$ but $p(x) > 0$, we apply elementwise $\varepsilon$-smoothing to both $p$ and $q$ with $\varepsilon = 10^{-8}$:

$$\tilde{r}(x) \;=\; \frac{r(x) + \varepsilon}{\sum_{x' \in \mathcal{X}}\big(r(x') + \varepsilon\big)} \quad \text{for } r \in \{p, q\},$$

and evaluate $D_{\mathrm{KL}}(\tilde{p}\|\tilde{q})$. This guarantees strictly positive probabilities and prevents $\log 0$.

## B.2  BN KL Divergence Calculation

The BN KL divergence is the standard KL divergence between two BNs that share the same structure (Koller & Friedman, 2009), decomposed into a weighted sum of local CPT divergences where the weights are the parent marginals $p(\mathrm{pa}_i)$, as shown in Figure 6. This decomposition does not eliminate the computational cost of computing the KL divergence: obtaining the parent marginals $p(\mathrm{pa}_i)$ requires exact inference, which is intractable in general for BNs with high treewidth. In our setting, the real-world BNs from bnRep have sufficiently low treewidth that variable elimination computes these marginals efficiently.

## B.3  CPT KL Divergence Calculation

In some of our experiments, we report a CPT KL divergence defined over local CPT rows instead of the entire BN. For each node $X_i$ with parent set $pa_i$ and parent configuration set $\mathcal{P}_i$, define the local divergence

$$d_{i,u} \;=\; D_{\mathrm{KL}}\big(p(X_i \mid pa_i{=}u) \,\|\, q(X_i \mid pa_i{=}u)\big), \quad u \in \mathcal{P}_i.$$

Our CPT KL divergence is the unweighted mean over all CPT rows in the BN:

$$\text{CPT KL divergence} \;=\; \frac{1}{\sum_{i=1}^{n} |\mathcal{P}_i|} \sum_{i=1}^{n} \sum_{u \in \mathcal{P}_i} d_{i,u}.$$

This metric treats every CPT row equally and is therefore suitable for subgroup analyses, e.g., by number of states or by number of parents. Unlike the BN KL divergence, the CPT KL divergence requires no inference over the network and it is computed directly from the CPT entries. In principle, the number of rows $|\mathcal{P}_i|$ can grow exponentially with the number of parents, but the real-world BNs in our experiments have small parent sets, keeping the tables compact and the computation efficient.

# C  Setup and Hyperparameters

## C.1  Large Language Models' Versions

In our experiments, we evaluated multiple state-of-the-art Large Language Models: GPT-4o and its mini variant (OpenAI et al., 2024) versions "gpt-4o-2024-11-20" and "gpt-4o-mini-2024-07-18", Claude 3.5 Sonnet (Anthropic, 2024) version "Claude 3.5 Sonnet 2024-10-22", Gemini 1.5 Pro (Gemini Team et al., 2024)

BN KL Divergence Decomposition into a Sum of Local KL Divergences

Let $p(\mathbf{x})$ and $q(\mathbf{x})$ be two BNs over the same variables $\{X_1, \ldots, X_n\}$ with common structure. Each factorizes as $p(\mathbf{x}) = \prod_{i=1}^{n} p\left(x_i \mid \mathrm{Pa}_i\right)$, $q(\mathbf{x}) = \prod_{i=1}^{n} q\left(x_i \mid \mathrm{Pa}_i\right)$, where $\mathrm{Pa}_i$ are the parents of $X_i$. We want to show:

$$D_{\mathrm{KL}}(p \,\|\, q) = \sum_{i=1}^{n} \sum_{\mathrm{pa}_i} p(\mathrm{pa}_i) \, D_{\mathrm{KL}}\big(p(X_i \mid \mathrm{pa}_i) \,\|\, q(X_i \mid \mathrm{pa}_i)\big).$$

**Derivation.**

$$D_{\mathrm{KL}}(p \,\|\, q) = \sum_{\mathbf{x}} p(\mathbf{x}) \log \frac{p(\mathbf{x})}{q(\mathbf{x})} = \sum_{\mathbf{x}} p(\mathbf{x}) \log \frac{\prod_{i=1}^{n} p(x_i \mid \mathrm{Pa}_i)}{\prod_{i=1}^{n} q(x_i \mid \mathrm{Pa}_i)} = \sum_{\mathbf{x}} p(\mathbf{x}) \sum_{i=1}^{n} \log \frac{p(x_i \mid \mathrm{Pa}_i)}{q(x_i \mid \mathrm{Pa}_i)}$$

$$= \sum_{i=1}^{n} \sum_{\mathbf{x}} p(\mathbf{x}) \log \frac{p(x_i \mid \mathrm{Pa}_i)}{q(x_i \mid \mathrm{Pa}_i)} = \sum_{i=1}^{n} \sum_{\mathrm{pa}_i} p(\mathrm{pa}_i) \sum_{x_i} p(x_i \mid \mathrm{pa}_i) \log \frac{p(x_i \mid \mathrm{pa}_i)}{q(x_i \mid \mathrm{pa}_i)}$$

$$= \sum_{i=1}^{n} \sum_{\mathrm{pa}_i} p(\mathrm{pa}_i) \, D_{\mathrm{KL}}\big(p(X_i \mid \mathrm{pa}_i) \,\|\, q(X_i \mid \mathrm{pa}_i)\big).$$

**Example.** Let $A, B, C \in \{0, 1\}$ within the network $A \to B \to C$. We compute $D_{\mathrm{KL}}(p\|q)$ as follows:

**1. Substitute the factorizations:**

$$D_{\mathrm{KL}}(p\|q) = \sum_{a,b,c} p(a,b,c) \log \frac{p(a,b,c)}{q(a,b,c)}. = \sum_{a,b,c} p(a,b,c) \log \frac{p(a)\,p(b \mid a)\,p(c \mid b)}{q(a)\,q(b \mid a)\,q(c \mid b)}.$$

**2. Separate logs:**

$$= \sum_{a,b,c} p(a,b,c) \left[ \log \frac{p(a)}{q(a)} + \log \frac{p(b \mid a)}{q(b \mid a)} + \log \frac{p(c \mid b)}{q(c \mid b)} \right].$$

**3. Split the sum:**

$$= \underbrace{\sum_{a,b,c} p(a,b,c) \log \frac{p(a)}{q(a)}}_{A} + \underbrace{\sum_{a,b,c} p(a,b,c) \log \frac{p(b \mid a)}{q(b \mid a)}}_{B} + \underbrace{\sum_{a,b,c} p(a,b,c) \log \frac{p(c \mid b)}{q(c \mid b)}}_{C}.$$

**4. Marginalize:**

$$A = \sum_{a} p(a) \log \frac{p(a)}{q(a)}, \quad B = \sum_{a} p(a) \sum_{b} p(b \mid a) \log \frac{p(b \mid a)}{q(b \mid a)}, \quad C = \sum_{b} p(b) \sum_{c} p(c \mid b) \log \frac{p(c \mid b)}{q(c \mid b)}.$$

**5. Recognize KL pieces and combine:**

$$D_{\mathrm{KL}}(p\|q) = D_{\mathrm{KL}}\big(p(A)\|q(A)\big) + \sum_{a} p(a) \, D_{\mathrm{KL}}\big(p(B \mid a)\|q(B \mid a)\big) + \sum_{b} p(b) \, D_{\mathrm{KL}}\big(p(C \mid b)\|q(C \mid b)\big).$$

Figure 6: KL divergence decomposition into local components for Bayesian Networks.

version "gemini-1.5-pro-002" and DeepSeek-V3 (DeepSeek-AI et al., 2024), version "DeepSeek-V3-0324". To test the reasoning models we used o3 and its mini variant (OpenAI, 2025), versions "o3-2025-04-16" and "o3-mini-2025-01-31", and DeepSeek-R1 (DeepSeek-AI et al., 2025), version "DeepSeek-R1-0528". All models were interfaced using the LangChain framework (Chase, 2022), ensuring consistent interaction.

## C.2 LLMs' Hyperparameters

A "temperature" of 0.1 was utilized to maintain minimal stochasticity in outputs. For reasoning models, the "reasoning effort" was set to *medium*. We initially explored the impact of sampling by performing up to five repeated samples per inference. However, we found that multiple samples did not meaningfully affect aggregate outcomes across the evaluated set of eighty Bayesian Networks, likely due to the low temperature setting. Consequently, given the number of LLMs and the dataset size, all subsequent experiments used a single sample to control cost. Output lengths were not constrained, allowing the models to elaborate their reasoning freely. In instances where models produced responses that deviated from the required format or where output text generation was interrupted midway, additional prompts were provided until valid responses were obtained.

## C.3 EDP's Hyperparameters

In our EDP formula, we have to select the hyperparameter, $\alpha$, in the following formula:

$$p_i = \frac{\alpha q_i + c_i}{\alpha + \sum_{j=1}^m c_j}, \qquad i = 1, \ldots, m.$$

where $q_i$ is the prior probability (from the LLM) and $c_i$ is the observed count for state $i$. Ideally, $\alpha$ is selected by testing various $\alpha$ values on a dev dataset of a downstream task. In our EDP experiments, where we are not testing a downstream task, we use a heuristic to select the $\alpha$. It is intuitive that as the number of data samples increases, the importance of priors decreases. As a result, we set $\alpha$ to be proportional to the inverse of the number of data samples. The same alpha is used for the uniform baseline, ensuring fairness. In our experiments, the results are very robust regarding alpha, and when we tested alpha of c/N where c is a multiplier, EDP consistently outperforms the uniform prior in all LLMs.

To assess the robustness of EDP to the choice of $\alpha$, we conducted a sensitivity analysis by varying the multiplier $c$ in $\alpha = c/N$, where $N$ is the number of data samples. We tested four values: $c \in \{0.5, 1, 2, 5\}$. The same $\alpha$ is always used for both EDP and the Uniform baseline to ensure a fair comparison. Figures 8 to 10 present the BN KL divergence distributions for each $\alpha$ setting. Across all four configurations, EDP consistently outperforms the Uniform prior at every sample size, confirming that the advantage of LLM-derived priors over uninformed priors is not an artifact of a particular $\alpha$ choice.

However, a poorly chosen $\alpha$ can degrade the performance of both EDP and the Uniform baseline. This is most visible with $c = 5$ (Figure 10), where giving too much weight to the prior relative to the data worsens results for both methods. For instance, EDP-3 and Uniform-3 are pushed further to the right in the sorted ordering compared to the default setting. Even in this unfavorable regime, EDP still outperforms the corresponding Uniform baseline, demonstrating the value of an informative prior even when $\alpha$ is suboptimal. The moderate settings ($c \in \{0.5, 1, 2\}$, Figures 7 to 9) all produce similar results, with EDP reliably improving over the Uniform prior across the full range of sample sizes. This stability confirms that EDP's improvements are robust and not sensitive to the exact value of $\alpha$, as long as it remains within a reasonable range.

For classification experiments using EDP, for a dataset with $N$ data samples, we set $\alpha = 0.5 \times N$ in the full-data regime. In the low-data regime, we select the optimal $\alpha$ from $0.5 \times N, 1.0 \times N, 2.0 \times N$ using the remaining training data as a development set. For classification tasks, we use different alphas because these involve realistic, not sampled, datasets where more data instances do not always help (e.g., Puffin reaches 100% accuracy with only 69 instances on MLE, while Pokemon reaches 62% with 999 examples). We use recommended values from the pgmpy library, and our method is compared against the best baselines. However, most models chose $0.5 \times N$ as the $\alpha$, making it the best default in the absence of additional information or a dev dataset.

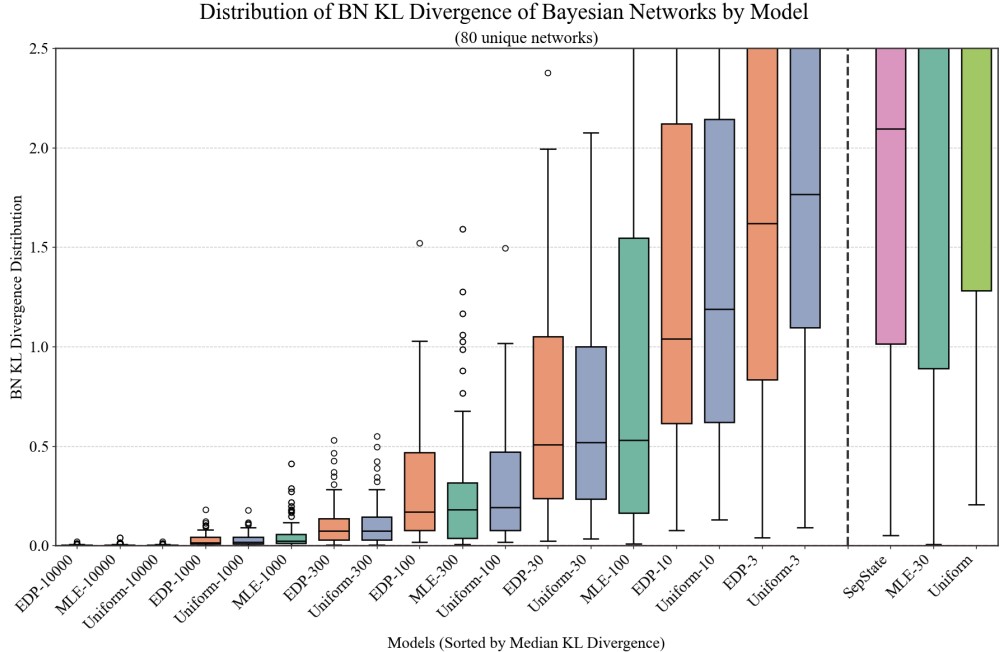

Figure 7: BN KL divergence with $\alpha = 0.5/N$. Results are comparable to the default, with EDP consistently outperforming the Uniform prior.

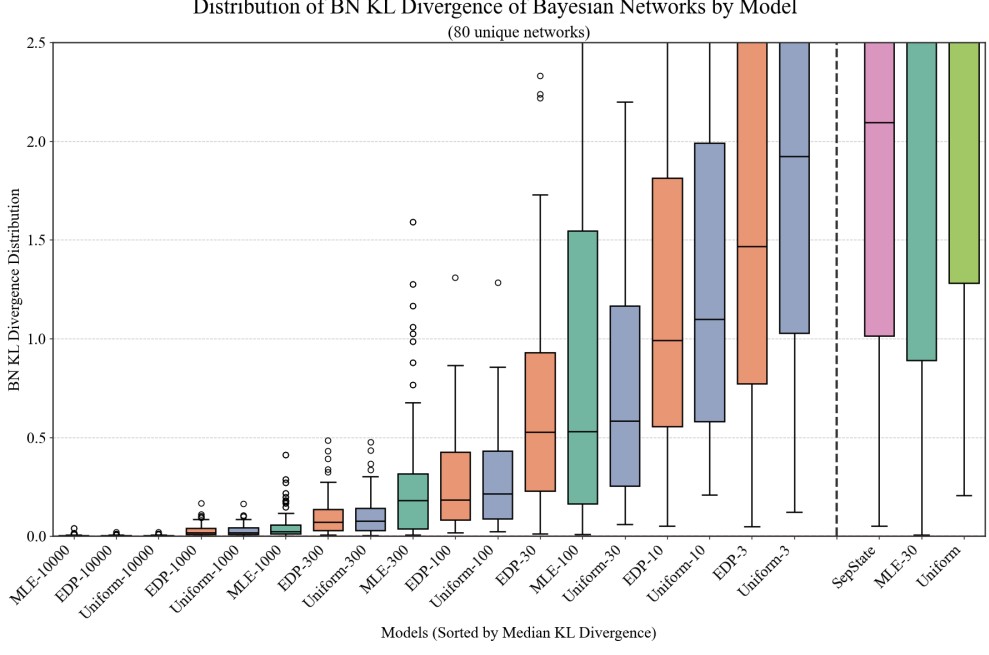

Figure 8: BN KL divergence with the default $\alpha = 1/N$. EDP outperforms the Uniform prior at every sample size.

## C.4 LLMs' Prompting Template for Token Probabilities

Figure 11 shows the prompt template we use to query the LLMs for token probability. The LLM is asked for only the most probable state name without any extra text. From the returned token scores for each

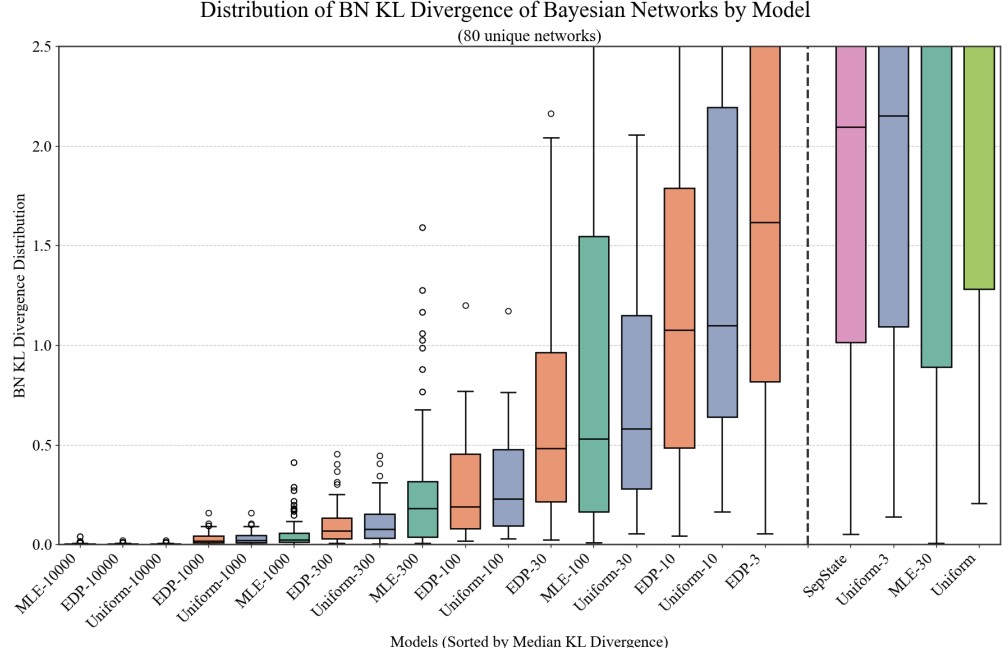

Figure 9: BN KL divergence with $\alpha = 2/N$. EDP maintains its advantage over the Uniform prior across all sample sizes.

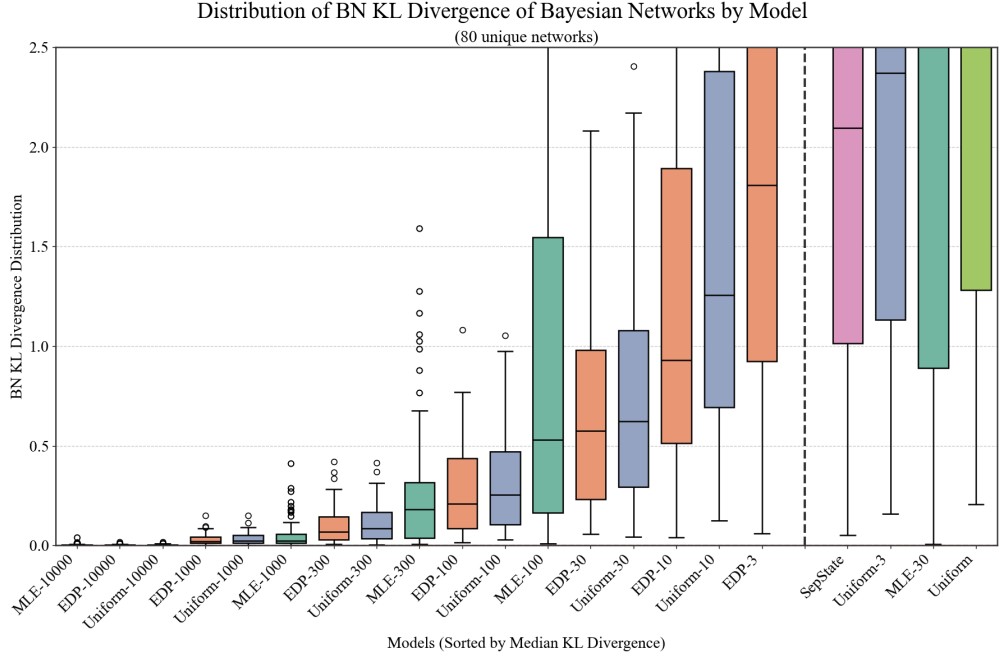

Figure 10: BN KL divergence with $\alpha = 5/N$. This over-weighting of the prior degrades both EDP and the Uniform baseline (e.g., EDP-3 and Uniform-3 are shifted rightward compared to the default). Nevertheless, EDP still outperforms the Uniform prior at every sample size.

candidate string, we compute the probabilities of the states of the nodes. If a node's state is not among the 20 returned candidate strings, we assign it a probability of 0. This occurs in only 12% of cases for

Figure 11: Prompt templates used for eliciting probabilistic responses from LLMs. The prompting structures for SepState, FullDist, and Extracting token probabilities are shown in the top, middle, and bottom panels, respectively.

GPT-4o and GPT-4o-mini. In our experiments, the probabilities of the returned strings typically sum to approximately 99%, justifying our assumption of a 0% probability for the absent states.

## D  Additional Experiments

### D.1  EDP's Impact on Downstream Tasks

Table 2 expands the classification study described in the main text by reporting additional results for DeepSeek-V3 and GPT-4o-mini and showing the variance in the results obtained in multiple runs. In the low-resource settings, the training cases are sampled from the training split, and results are averaged over five random runs. The $\pm$ values in the 20 and 10 sample columns denote the empirical standard deviation across the five runs. For the LLM-only reference column, we use SepState with GPT-4o and GPT-4o-mini and FullDist with DeepSeek-V3. The corresponding EDP columns use the same extraction scheme as their LLM prior.

To assess the statistical significance of the performance differences between the EDP and MLE methods, we pooled the evaluation results from our three Large Language Models (GPT-4o, GPT-4o-mini, and DeepSeek-V3) by concatenating their respective Macro $F_1$ scores. We utilized a one-sided Wilcoxon signed-rank test to determine if the performance of the EDP method was significantly greater than that of the MLE method. To ensure robustness against tied ranks, we employed Pratt's method to handle zero-difference pairs, considering results statistically significant at the $p < 0.05$ threshold. As shown in Table 3, the results are statistically significant for all the low-data regimes in both BN structures.

### D.2  EDP with Other LLMs and Sampling Methods

In this paper, we demonstrated that EDP improves probability estimates as measured by KL divergence, using GPT-4o. However, this trend is consistent across other LLMs as well. Figures 12, 13, and 14 illustrate EDP's results for DeepSeek-V3, Gemini 1.5 Pro, and Claude 3.5 Sonnet, respectively.

In our main experiments, we performed forward sampling on the entire BN. However, EDP consistently improves results under alternative sampling methods. Figure 15 shows EDP's performance when sampling # data points per CPT row. Similarly, Figure 16 presents the results when using a related sampling strategy, where each CPT row is sampled $\# \times n$ times, with $n$ representing the number of states of the node. These results prove that EDP's improvements are independent of the sampling method.

**GPT-4o**

| Dataset | | Hill Climbing | | | | | | | Naive Bayes | | | | | |
|---|---|---|---|---|---|---|---|---|---|---|---|---|---|---|
| | | Full Data | | 20 Samples | | 10 Samples | | | Full Data | | 20 Samples | | 10 Samples | |
| | SS | MLE | EDP | MLE | EDP | MLE | EDP | SS | MLE | EDP | MLE | EDP | MLE | EDP |
| HV84* | .24 | **.94** | .91 | $.87_{\pm.05}$ | $.80_{\pm.16}$ | $.78_{\pm.10}$ | $\mathbf{.85}_{\pm.13}$ | .40 | **.94** | .91 | $\mathbf{.92}_{\pm.01}$ | $.90_{\pm.02}$ | $\mathbf{.91}_{\pm.02}$ | $.87_{\pm.01}$ |
| PhDA* | .41 | .38 | **.41** | $.35_{\pm.05}$ | $\mathbf{.41}_{\pm.00}$ | $.34_{\pm.04}$ | $\mathbf{.41}_{\pm.01}$ | .45 | **.44** | .44 | $.36_{\pm.03}$ | $\mathbf{.41}_{\pm.01}$ | $.34_{\pm.04}$ | $\mathbf{.41}_{\pm.03}$ |
| Pokemon | .23 | **.62** | .62 | $\mathbf{.62}_{\pm.00}$ | $\mathbf{.62}_{\pm.00}$ | $.54_{\pm.12}$ | $\mathbf{.54}_{\pm.17}$ | .48 | **.62** | .60 | $.59_{\pm.05}$ | $\mathbf{.60}_{\pm.01}$ | $.53_{\pm.04}$ | $.54_{\pm.08}$ |
| Titanic | .42 | **.42** | .42 | $.42_{\pm.00}$ | $.42_{\pm.00}$ | $.42_{\pm.00}$ | $.42_{\pm.00}$ | .57 | .11 | **.55** | $.22_{\pm.14}$ | $\mathbf{.57}_{\pm.00}$ | $.42_{\pm.15}$ | $\mathbf{.56}_{\pm.01}$ |
| CAD1 | .87 | **.83** | .83 | $.74_{\pm.05}$ | $\mathbf{.85}_{\pm.02}$ | $.63_{\pm.08}$ | $\mathbf{.84}_{\pm.04}$ | .77 | .81 | **.85** | $.83_{\pm.04}$ | $\mathbf{.86}_{\pm.03}$ | $.80_{\pm.06}$ | $.84_{\pm.05}$ |
| CAD2 | .76 | .65 | **.76** | $\mathbf{.76}_{\pm.00}$ | $\mathbf{.76}_{\pm.00}$ | $.60_{\pm.14}$ | $\mathbf{.76}_{\pm.00}$ | .50 | **.86** | .79 | $.73_{\pm.08}$ | $\mathbf{.78}_{\pm.03}$ | $.77_{\pm.10}$ | $\mathbf{.78}_{\pm.06}$ |
| Covid | .74 | .71 | **.73** | $.70_{\pm.03}$ | $\mathbf{.72}_{\pm.01}$ | $.72_{\pm.01}$ | $\mathbf{.72}_{\pm.00}$ | .72 | .71 | **.72** | $.71_{\pm.00}$ | $\mathbf{.72}_{\pm.01}$ | $.69_{\pm.03}$ | $\mathbf{.72}_{\pm.00}$ |
| Puffin | .63 | **1.0** | .93 | $\mathbf{.99}_{\pm.03}$ | $.91_{\pm.03}$ | $\mathbf{.97}_{\pm.04}$ | $.91_{\pm.03}$ | .78 | **.93** | .85 | $\mathbf{.90}_{\pm.07}$ | $.88_{\pm.07}$ | $.87_{\pm.06}$ | $\mathbf{.88}_{\pm.04}$ |
| Traject* | .87 | **.87** | .87 | $.75_{\pm.06}$ | $\mathbf{.86}_{\pm.01}$ | $.68_{\pm.07}$ | $\mathbf{.86}_{\pm.01}$ | .80 | **.87** | .87 | $\mathbf{.86}_{\pm.00}$ | $.86_{\pm.01}$ | $.85_{\pm.03}$ | $\mathbf{.86}_{\pm.02}$ |
| Average | .57 | .71 | **.72** | $.69_{\pm.03}$ | $\mathbf{.71}_{\pm.03}$ | $.63_{\pm.07}$ | $\mathbf{.70}_{\pm.05}$ | .61 | .70 | **.73** | $.68_{\pm.05}$ | $\mathbf{.73}_{\pm.02}$ | $.69_{\pm.06}$ | $\mathbf{.72}_{\pm.03}$ |

**DeepSeek-V3**

| Dataset | | Hill Climbing | | | | | | | Naive Bayes | | | | | |
|---|---|---|---|---|---|---|---|---|---|---|---|---|---|---|
| | | Full Data | | 20 Samples | | 10 Samples | | | Full Data | | 20 Samples | | 10 Samples | |
| | FD | MLE | EDP | MLE | EDP | MLE | EDP | FD | MLE | EDP | MLE | EDP | MLE | EDP |
| HV84* | .93 | **.94** | .93 | $.87_{\pm.06}$ | $\mathbf{.94}_{\pm.03}$ | $.79_{\pm.11}$ | $\mathbf{.92}_{\pm.00}$ | .51 | **.94** | .92 | $\mathbf{.92}_{\pm.01}$ | $.88_{\pm.02}$ | $\mathbf{.91}_{\pm.02}$ | $.89_{\pm.01}$ |
| PhDA* | .38 | **.38** | .38 | $.35_{\pm.05}$ | $\mathbf{.38}_{\pm.00}$ | $.34_{\pm.04}$ | $\mathbf{.38}_{\pm.00}$ | .34 | **.44** | .41 | $.36_{\pm.03}$ | $\mathbf{.40}_{\pm.03}$ | $.34_{\pm.04}$ | $\mathbf{.40}_{\pm.05}$ |
| Pokemon | .62 | **.62** | .62 | $\mathbf{.62}_{\pm.00}$ | $.59_{\pm.08}$ | $.54_{\pm.12}$ | $\mathbf{.59}_{\pm.08}$ | .43 | **.62** | .62 | $.59_{\pm.05}$ | $\mathbf{.61}_{\pm.01}$ | $.53_{\pm.04}$ | $\mathbf{.59}_{\pm.05}$ |
| Titanic | .42 | **.42** | .42 | $\mathbf{.42}_{\pm.00}$ | $\mathbf{.42}_{\pm.00}$ | $\mathbf{.42}_{\pm.00}$ | $\mathbf{.42}_{\pm.00}$ | .57 | .11 | **.57** | $.22_{\pm.14}$ | $\mathbf{.57}_{\pm.00}$ | $.42_{\pm.15}$ | $\mathbf{.56}_{\pm.01}$ |
| CAD1 | .89 | .83 | **.88** | $.74_{\pm.05}$ | $\mathbf{.86}_{\pm.04}$ | $.63_{\pm.08}$ | $\mathbf{.85}_{\pm.03}$ | .81 | .81 | **.88** | $.83_{\pm.04}$ | $\mathbf{.85}_{\pm.03}$ | $.80_{\pm.06}$ | $\mathbf{.83}_{\pm.05}$ |
| CAD2 | .76 | .65 | **.76** | $.57_{\pm.16}$ | $\mathbf{.76}_{\pm.00}$ | $.57_{\pm.11}$ | $\mathbf{.76}_{\pm.00}$ | .33 | **.86** | .79 | $\mathbf{.73}_{\pm.08}$ | $.73_{\pm.12}$ | $.77_{\pm.10}$ | $\mathbf{.79}_{\pm.00}$ |
| Covid | .72 | .71 | **.73** | $.70_{\pm.03}$ | $\mathbf{.71}_{\pm.03}$ | $\mathbf{.72}_{\pm.01}$ | $.70_{\pm.03}$ | .72 | .71 | **.73** | $.71_{\pm.00}$ | $\mathbf{.72}_{\pm.01}$ | $.69_{\pm.03}$ | $\mathbf{.72}_{\pm.00}$ |
| Puffin | .33 | **1.0** | .93 | $\mathbf{.99}_{\pm.03}$ | $.93_{\pm.00}$ | $\mathbf{.97}_{\pm.04}$ | $.91_{\pm.03}$ | .48 | **.93** | .85 | $\mathbf{.90}_{\pm.07}$ | $.88_{\pm.07}$ | $\mathbf{.87}_{\pm.06}$ | $.87_{\pm.06}$ |
| Traject* | .59 | **.87** | .87 | $.75_{\pm.06}$ | $\mathbf{.84}_{\pm.03}$ | $.68_{\pm.07}$ | $\mathbf{.82}_{\pm.03}$ | .69 | **.87** | .81 | $\mathbf{.86}_{\pm.00}$ | $.82_{\pm.02}$ | $\mathbf{.85}_{\pm.03}$ | $.82_{\pm.04}$ |
| Average | .63 | .71 | **.72** | $.67_{\pm.05}$ | $\mathbf{.71}_{\pm.02}$ | $.63_{\pm.06}$ | $\mathbf{.71}_{\pm.02}$ | .54 | .70 | **.73** | $.68_{\pm.05}$ | $\mathbf{.72}_{\pm.03}$ | $.69_{\pm.06}$ | $\mathbf{.72}_{\pm.03}$ |

**GPT-4o-mini**

| Dataset | | Hill Climbing | | | | | | | Naive Bayes | | | | | |
|---|---|---|---|---|---|---|---|---|---|---|---|---|---|---|
| | | Full Data | | 20 Samples | | 10 Samples | | | Full Data | | 20 Samples | | 10 Samples | |
| | SS | MLE | EDP | MLE | EDP | MLE | EDP | SS | MLE | EDP | MLE | EDP | MLE | EDP |
| HV84* | .05 | **.94** | .82 | $\mathbf{.87}_{\pm.06}$ | $.68_{\pm.14}$ | $\mathbf{.79}_{\pm.11}$ | $.65_{\pm.17}$ | .30 | **.94** | .93 | $.86_{\pm.01}$ | $\mathbf{.91}_{\pm.02}$ | $\mathbf{.91}_{\pm.02}$ | $.90_{\pm.03}$ |
| PhDA* | .40 | .38 | **.41** | $.35_{\pm.05}$ | $\mathbf{.38}_{\pm.04}$ | $.34_{\pm.04}$ | $\mathbf{.38}_{\pm.04}$ | .31 | **.44** | .43 | $.36_{\pm.03}$ | $\mathbf{.40}_{\pm.02}$ | $.34_{\pm.04}$ | $\mathbf{.39}_{\pm.02}$ |
| Pokemon | .36 | **.62** | .62 | $\mathbf{.62}_{\pm.00}$ | $\mathbf{.62}_{\pm.00}$ | $.54_{\pm.12}$ | $.51_{\pm.17}$ | .54 | **.62** | .59 | $\mathbf{.59}_{\pm.05}$ | $.53_{\pm.07}$ | $\mathbf{.53}_{\pm.04}$ | $.49_{\pm.08}$ |
| Titanic | .57 | **.42** | .42 | $.42_{\pm.00}$ | $.42_{\pm.00}$ | $.42_{\pm.00}$ | $\mathbf{.42}_{\pm.00}$ | .57 | .11 | **.57** | $.22_{\pm.14}$ | $\mathbf{.57}_{\pm.00}$ | $.42_{\pm.15}$ | $\mathbf{.56}_{\pm.01}$ |
| CAD1 | .71 | **.83** | .83 | $.74_{\pm.05}$ | $\mathbf{.82}_{\pm.02}$ | $.63_{\pm.08}$ | $\mathbf{.82}_{\pm.02}$ | .60 | **.81** | .73 | $\mathbf{.83}_{\pm.04}$ | $.80_{\pm.05}$ | $\mathbf{.80}_{\pm.06}$ | $.83_{\pm.02}$ |
| CAD2 | .19 | **.65** | .65 | $.57_{\pm.16}$ | $\mathbf{.65}_{\pm.00}$ | $\mathbf{.57}_{\pm.11}$ | $.57_{\pm.11}$ | .51 | **.86** | .79 | $.73_{\pm.08}$ | $\mathbf{.84}_{\pm.03}$ | $.77_{\pm.10}$ | $\mathbf{.86}_{\pm.05}$ |
| Covid | .72 | **.71** | .71 | $.70_{\pm.03}$ | $\mathbf{.71}_{\pm.00}$ | $\mathbf{.72}_{\pm.01}$ | $.71_{\pm.00}$ | .54 | .71 | **.71** | $\mathbf{.71}_{\pm.00}$ | $.71_{\pm.00}$ | $.69_{\pm.03}$ | $\mathbf{.71}_{\pm.00}$ |
| Puffin | .59 | **1.0** | .93 | $\mathbf{.99}_{\pm.03}$ | $.91_{\pm.03}$ | $\mathbf{.97}_{\pm.04}$ | $.91_{\pm.03}$ | .48 | **.93** | .85 | $.90_{\pm.07}$ | $\mathbf{.91}_{\pm.03}$ | $\mathbf{.87}_{\pm.06}$ | $.87_{\pm.03}$ |
| Traject* | .80 | **.87** | .87 | $.81_{\pm.03}$ | $\mathbf{.84}_{\pm.03}$ | $.70_{\pm.06}$ | $\mathbf{.84}_{\pm.03}$ | .59 | **.87** | .85 | $\mathbf{.86}_{\pm.00}$ | $.82_{\pm.02}$ | $\mathbf{.85}_{\pm.03}$ | $.86_{\pm.01}$ |
| Average | .52 | **.71** | .69 | $\mathbf{.67}_{\pm.05}$ | $\mathbf{.67}_{\pm.03}$ | $.63_{\pm.06}$ | $\mathbf{.64}_{\pm.06}$ | .52 | .70 | **.73** | $.67_{\pm.05}$ | $\mathbf{.72}_{\pm.03}$ | $.69_{\pm.06}$ | $\mathbf{.71}_{\pm.03}$ |

Table 2: Results of Table 1 in more detail for DeepSeek-V3 and GPT-4o-mini, including the standard deviation for the five runs.

|  | Sample Size Condition | | |
|---|---|---|---|
|  | Full Data | 20 Samples | 10 Samples |
| Hill Climbing | 0.4227 | **0.0408** | **0.0130** |
| Naive Bayes | 0.9114 | **0.0482** | **0.0019** |

Table 3: Statistical Significance (*p*-values) across various data regimes and BN structures.

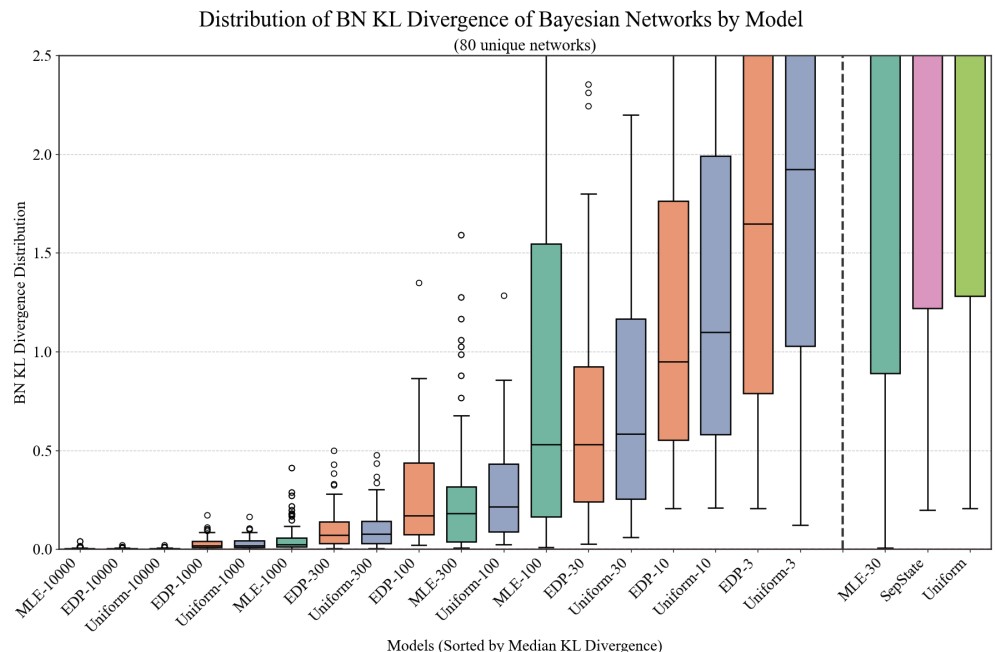

Figure 12: Boxplot of distribution of BN KL divergence over eighty networks, contrasting models with DeepSeek-V3 priors (EDP-#), uniform priors (Uniform-#), and data-only estimates (MLE-#). The numeral after the "-" denotes the sample size.

### D.3 SepState and FullDist with CPT KL Divergence

Figure 17 reports CPT KL for the methods discussed in the main paper. Consistent with the BN KL divergence, we observe that the relative ordering of methods is unchanged, meaning that the SepState performs the best among all baselines. Here, however, FullDist performs worse than the Uniform baseline in most LLMs.

## E  Varying Number of Parents and States

Figures 18 through 22 illustrate the CPT KL divergence for nodes with 2 to 6 states, respectively. Figures 23 through 30 depict the CPT KL divergence for nodes with 0 to 7 parents, respectively. These figures help to show the points made in the discussion section about the capability of LLMs in handling nodes with various state sizes and parents.

## F  BN KL Divergence by Domain Area

Figures 31 through 43 present the BN KL divergence distributions broken down by the domain area of each Bayesian Network. These per-domain results complement the aggregated analysis in the main text and provide finer-grained insight into how different LLMs perform across specialized fields. Models are sorted by

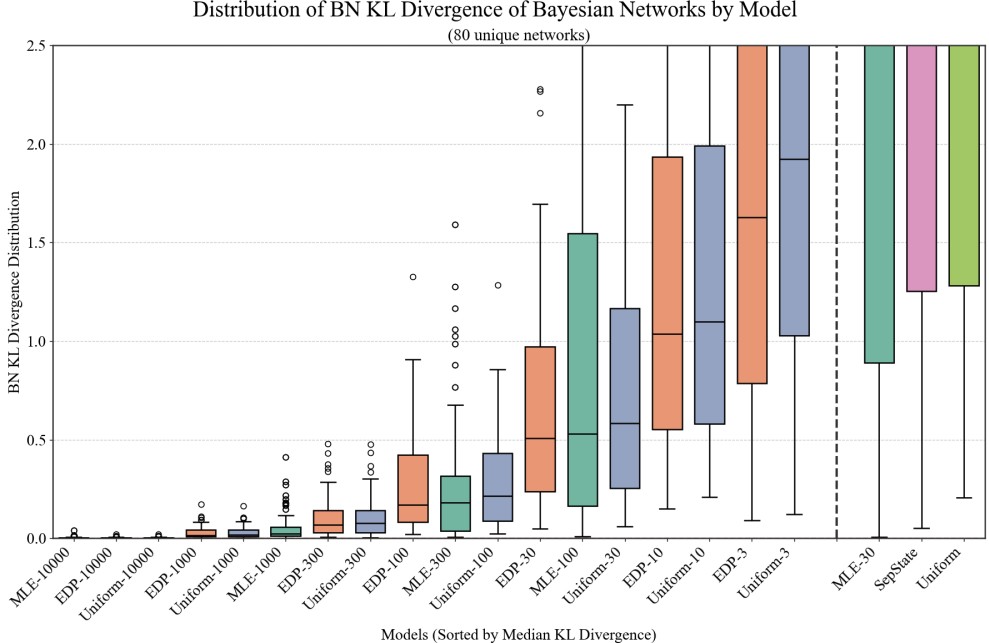

Figure 13: Boxplot of distribution of BN KL divergence over eighty networks, contrasting models with Gemini 1.5 Pro priors (EDP-#), uniform priors (Uniform-#), and data-only estimates (MLE-#). The numeral after the "-" denotes the sample size.

median KL divergence within each domain. We note that domains with just a few BNs offer limited statistical power and observed trends in these domains should be regarded as suggestive rather than definitive. Still, these figures show the difference in the knowledge of LLMs in various domains.

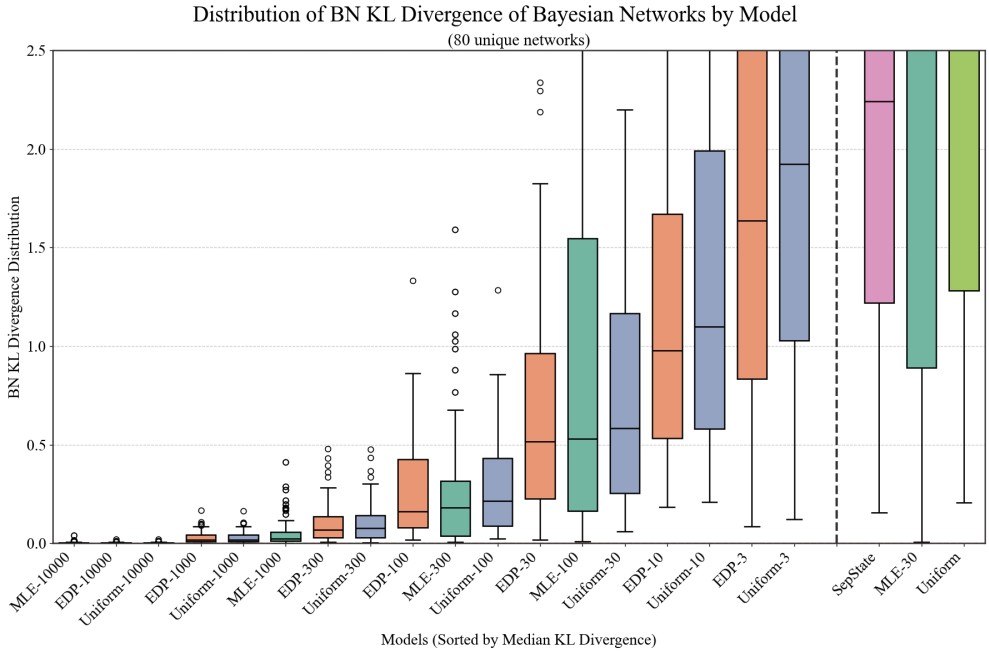

Figure 14: Boxplot of distribution of BN KL divergence over eighty networks, contrasting models with Claude 3.5 Sonnet priors (EDP-#), uniform priors (Uniform-#), and data-only estimates (MLE-#). The numeral after the "-" denotes the sample size.

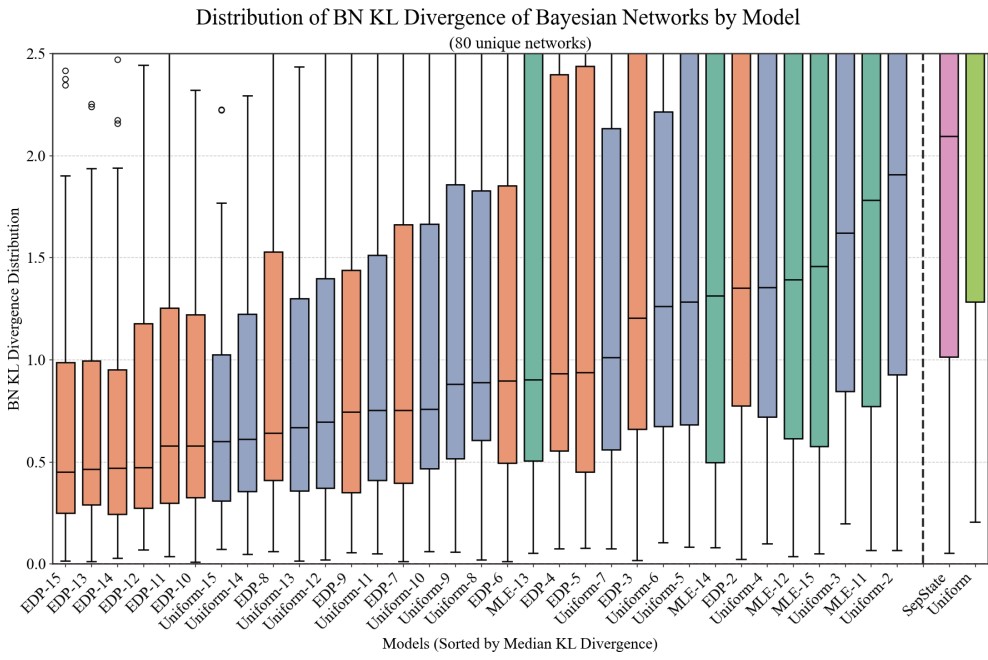

Figure 15: Boxplot of distribution of BN KL divergence over eighty networks, contrasting models with GPT-4o priors (EDP-#), uniform priors (Uniform-#), and data-only estimates (MLE-#). The numeral after the "-" denotes the sample size. In this figure, the data for each row of the CPT is sampled by # instances.

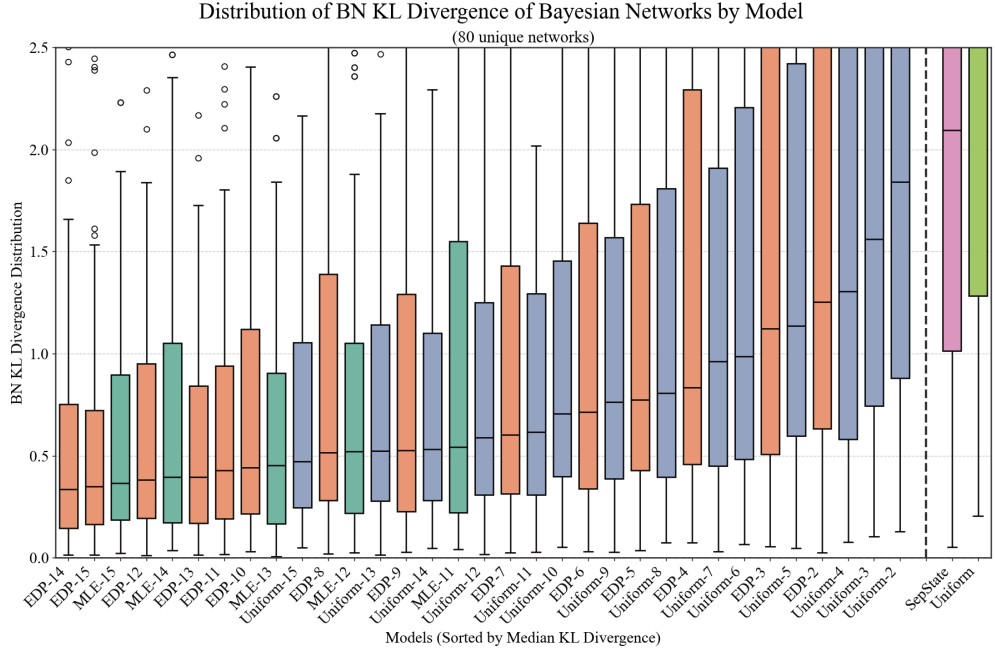

Figure 16: Boxplot of distribution of BN KL divergence over eighty networks, contrasting models with GPT-4o priors (EDP-#), uniform priors (Uniform-#), and data-only estimates (MLE-#). The numeral after the "-" denotes the sample size. In this figure, the data for each row of the CPT is sampled by $\# \times n$ times, with $n$ representing the number of states of the node.

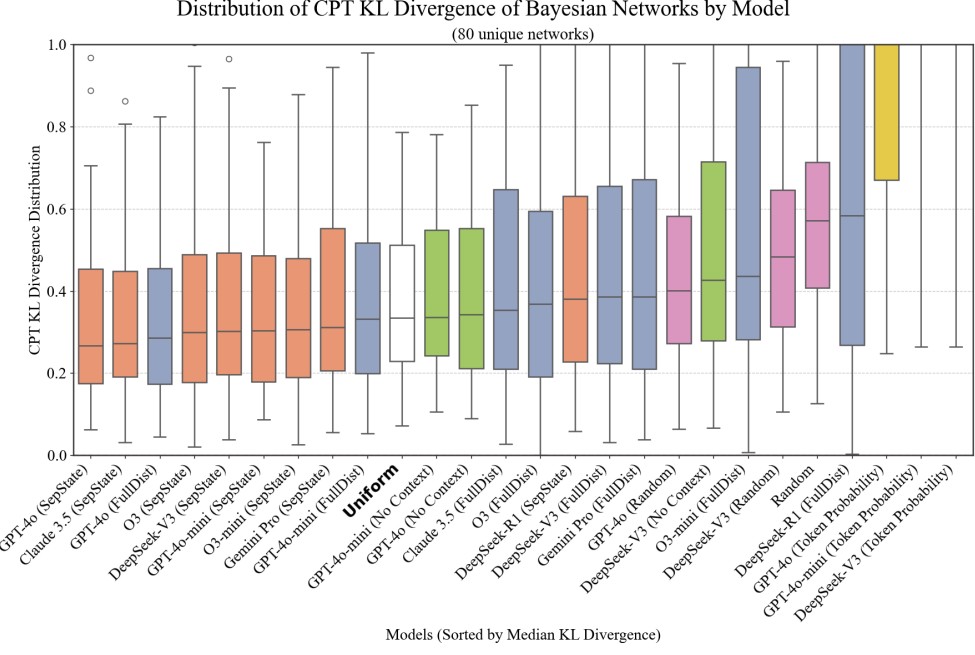

Figure 17: Boxplot showing the distribution of CPT KL divergence values across eighty unique BNs for various models, sorted by their median KL divergence. Lower values indicate better alignment with ground-truth CPTs.

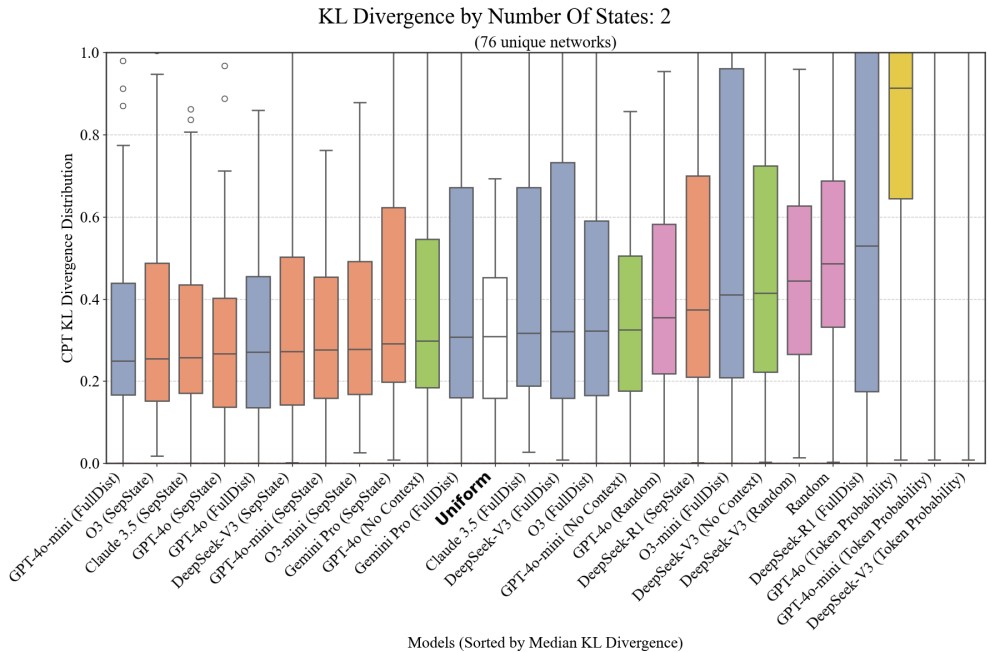

Figure 18: CPT KL divergence for nodes with 2 states.

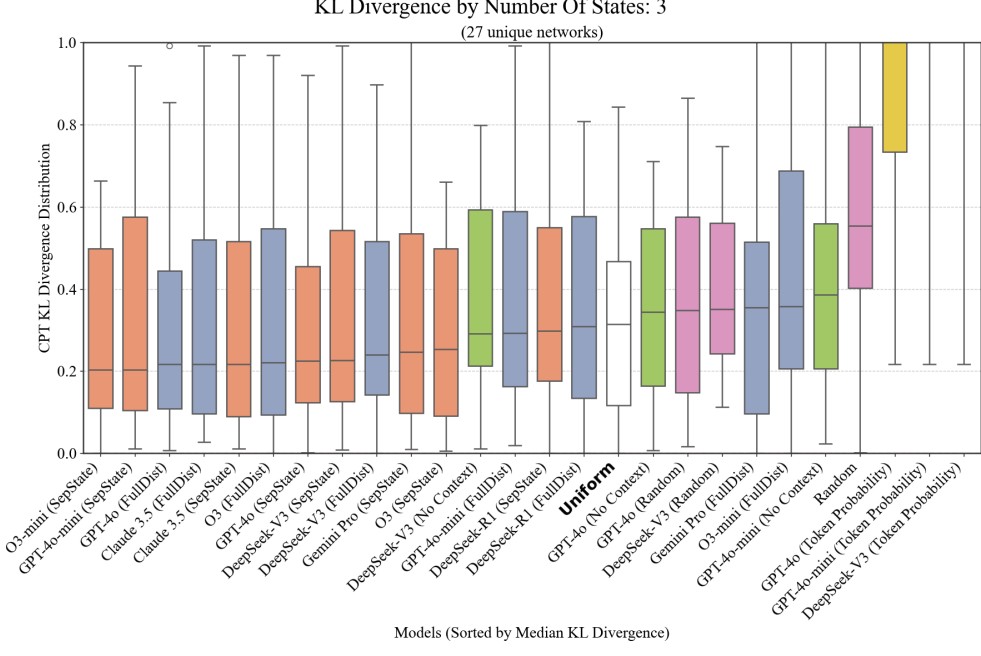

Figure 19: CPT KL divergence for nodes with 3 states.

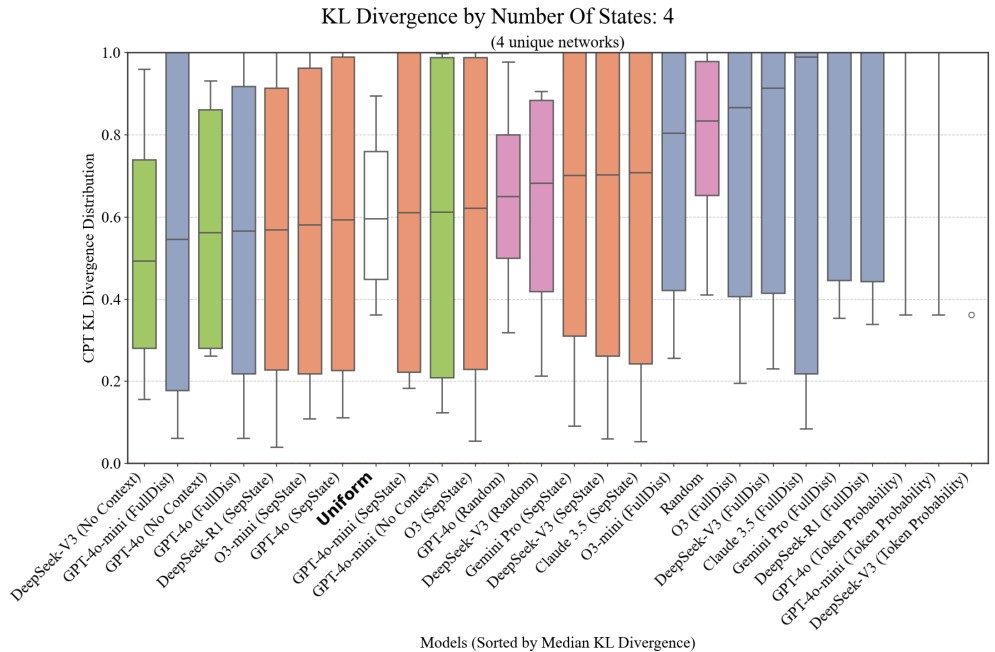

Figure 20: CPT KL divergence for nodes with 4 states.

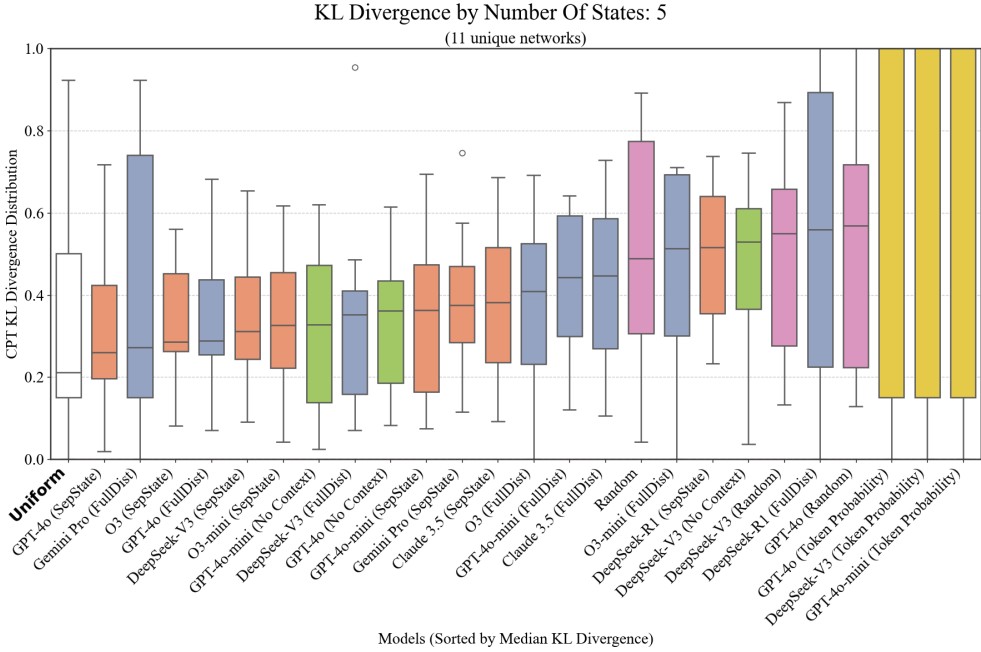

Figure 21: CPT KL divergence for nodes with 5 states.

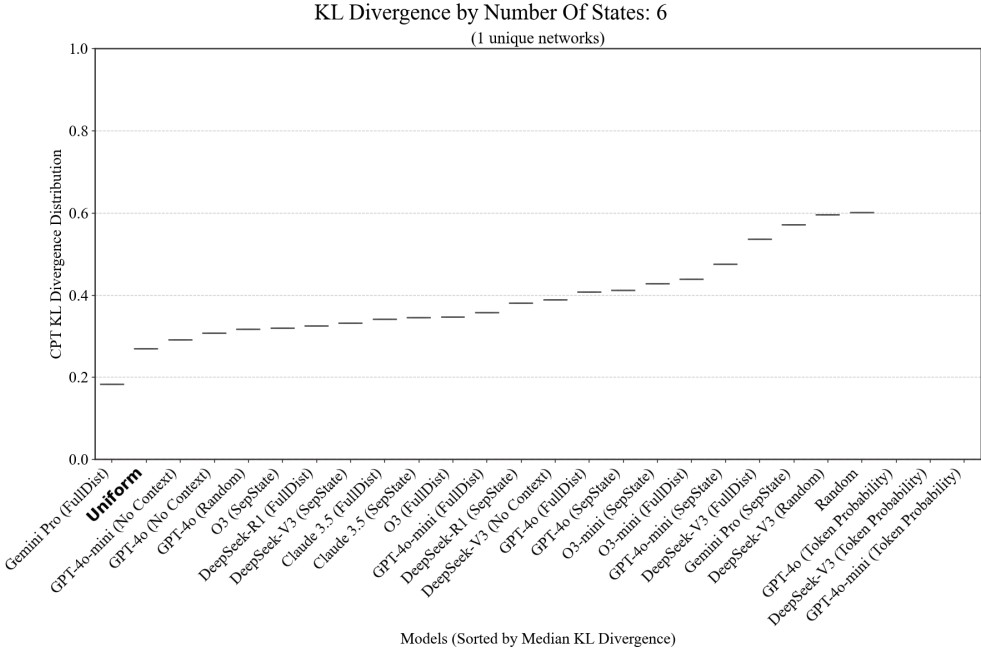

Figure 22: CPT KL divergence for nodes with 6 states.

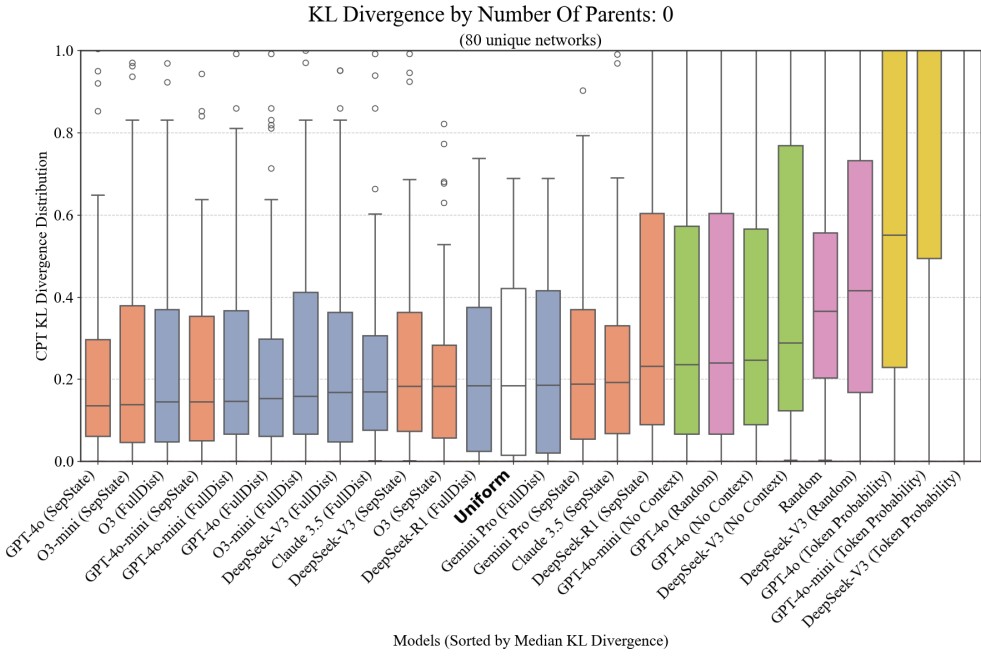

Figure 23: CPT KL divergence for nodes with 0 parents.

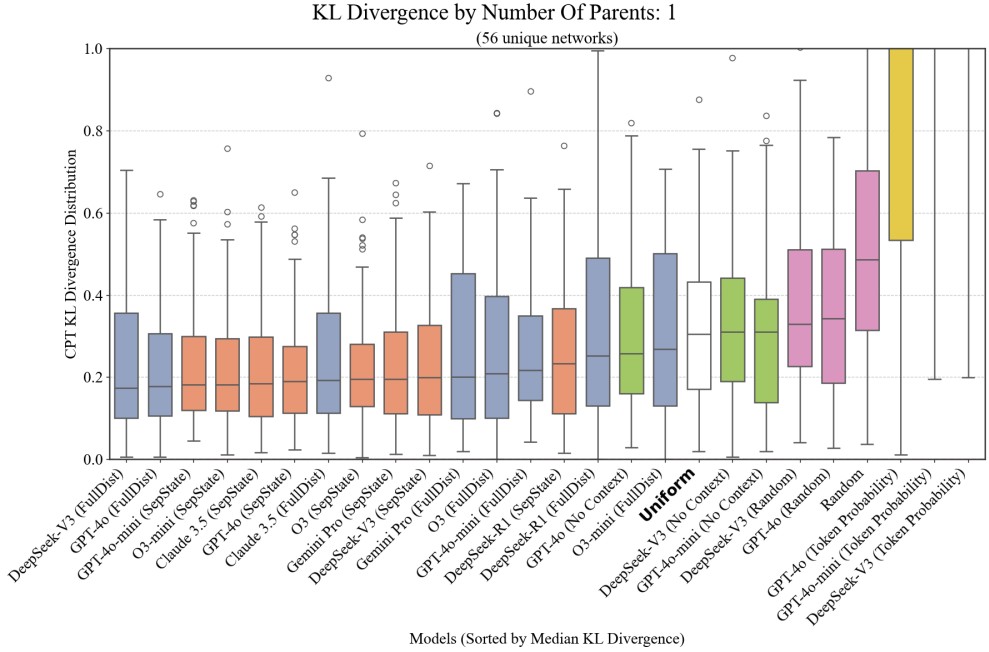

Figure 24: CPT KL divergence for nodes with 1 parent.

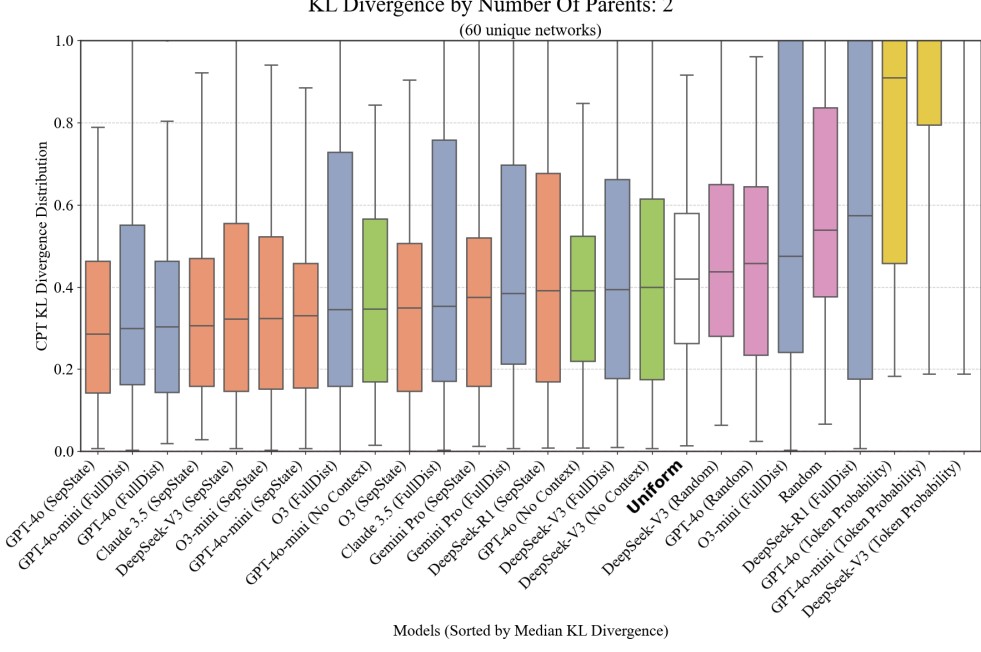

Figure 25: CPT KL divergence for nodes with 2 parents.

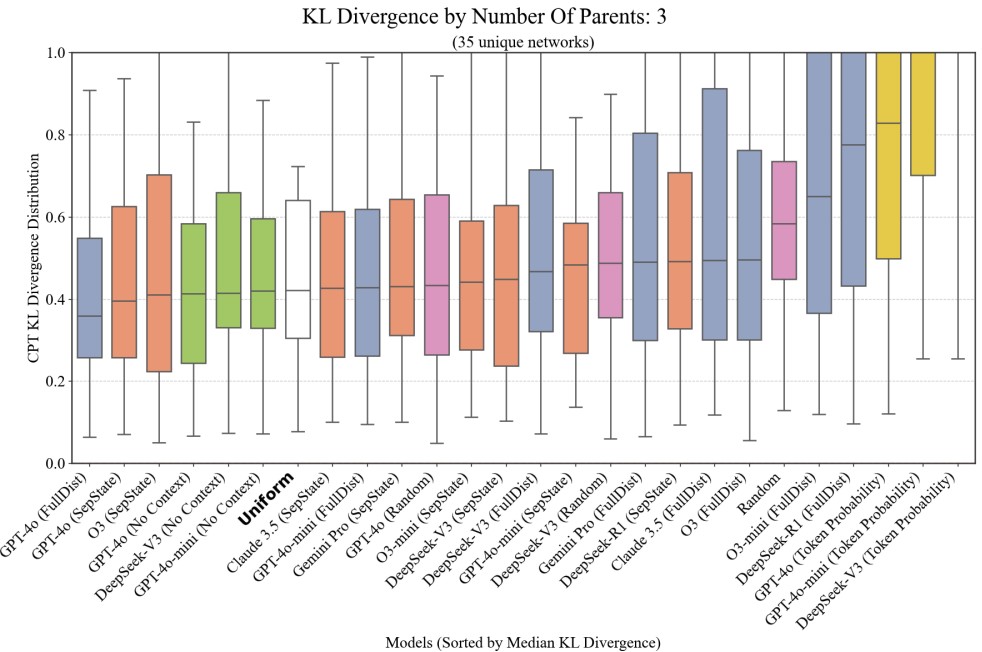

Figure 26: CPT KL divergence for nodes with 3 parents.

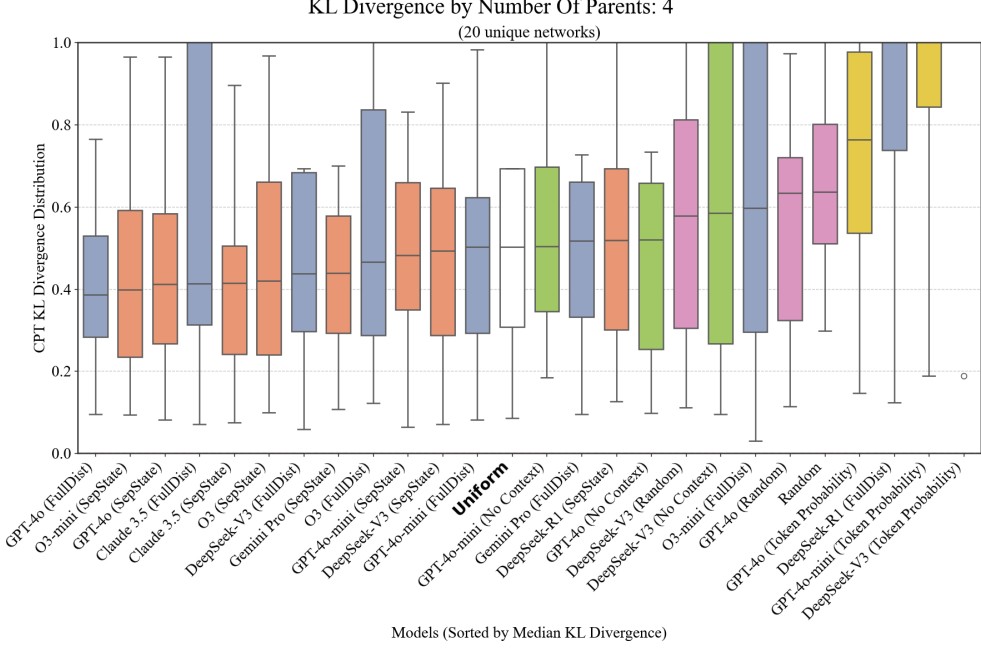

Figure 27: CPT KL divergence for nodes with 4 parents.

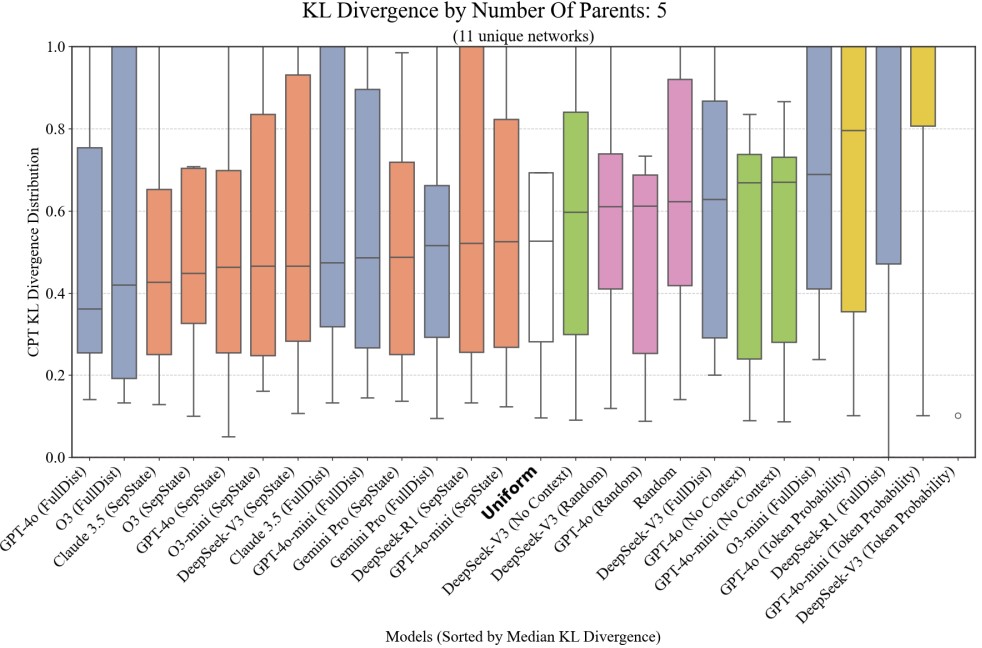

Figure 28: CPT KL divergence for nodes with 5 parents.

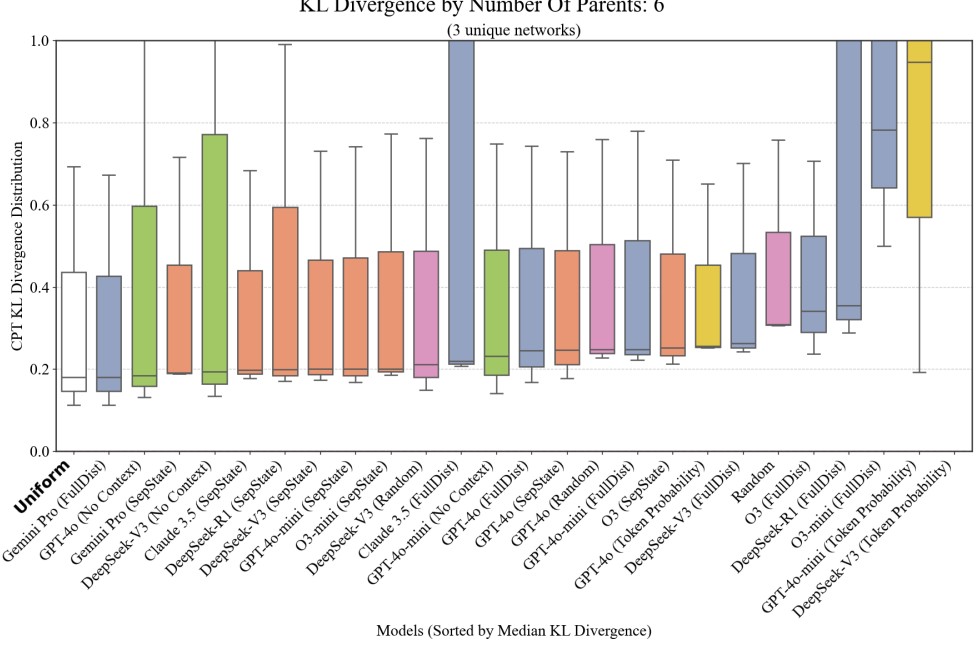

Figure 29: CPT KL divergence for nodes with 6 parents.

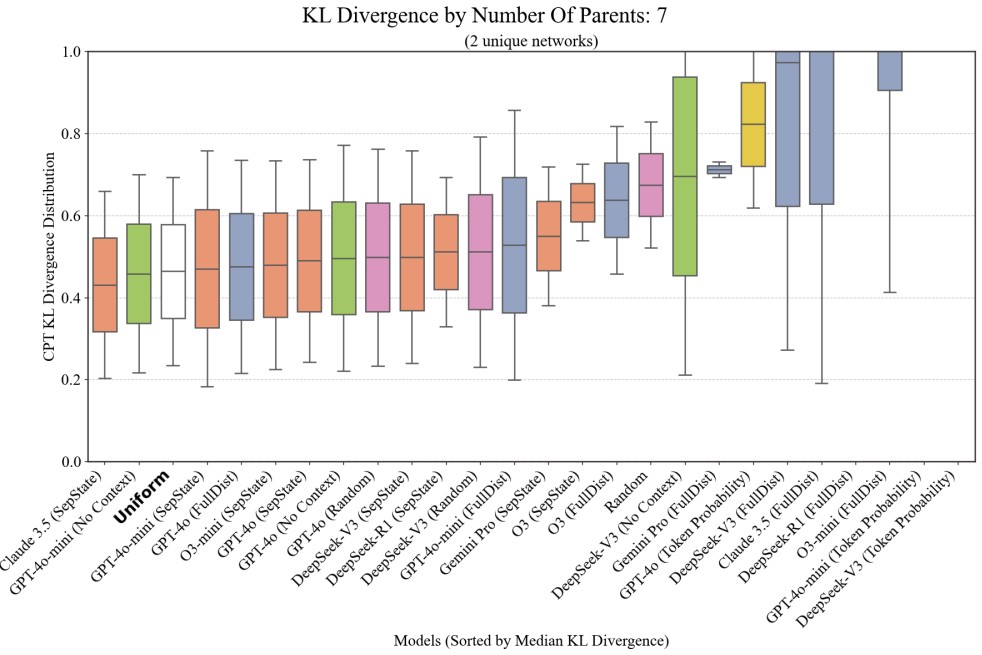

Figure 30: CPT KL divergence for nodes with 7 parents.

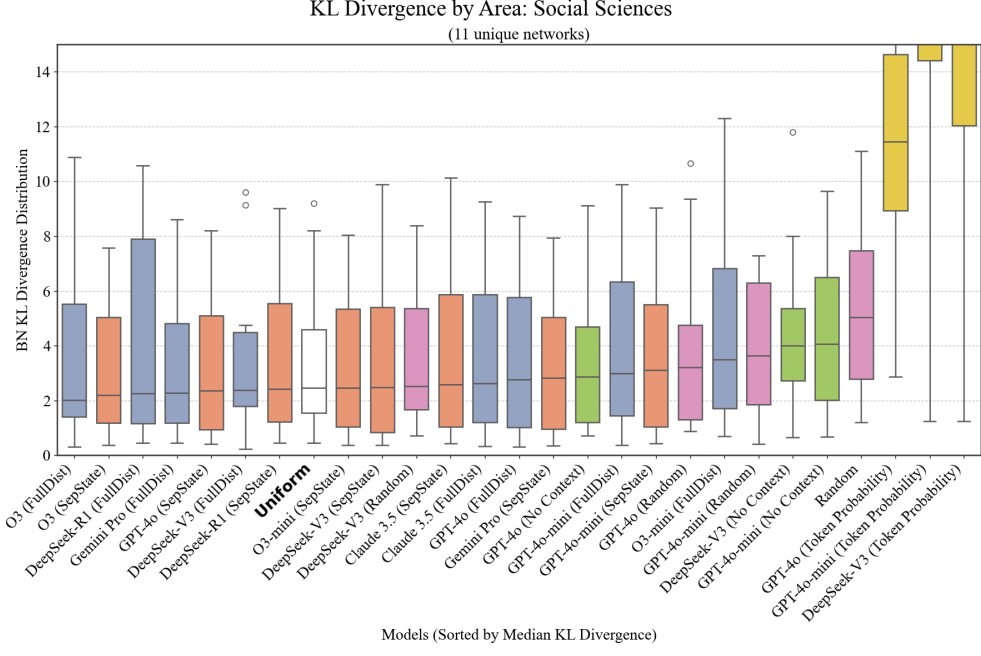

Figure 31: BN KL divergence by area: Social Sciences. With 11 BNs, this domain provides a reasonably sized sample for comparison. Most models are struggling in this field. o3 (FullDist) achieves the lowest median KL divergence, followed closely by o3 (SepState) and DeepSeek-R1 (FullDist). GPT-4o (SepState) still beats the Uniform baseline.

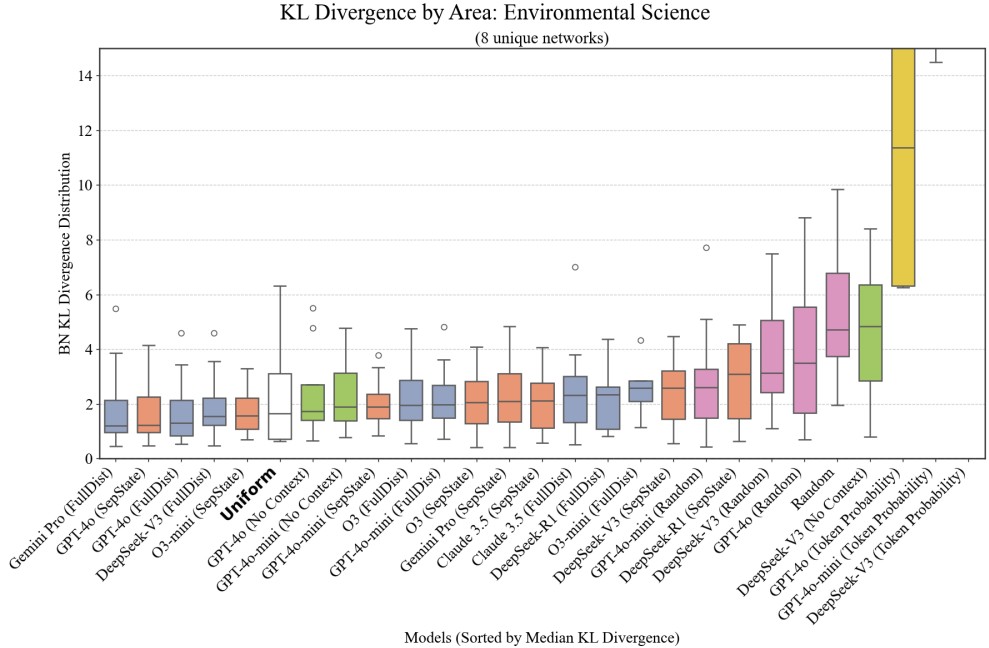

Figure 32: BN KL divergence by area: Environmental Science. Most models are struggling in this field. Gemini 1.5 Pro (FullDist) achieves the lowest median KL divergence, with GPT-4o (SepState) and GPT-4o (FullDist) close behind.

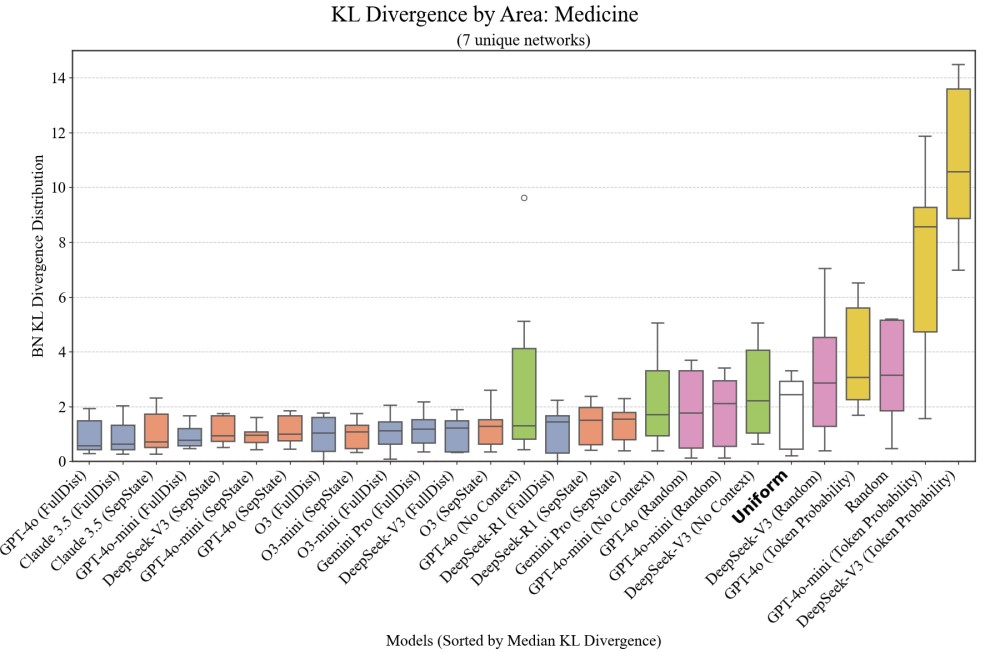

Figure 33: BN KL divergence by area: Medicine. Most models beat the Uniform baseline even without context, suggesting that they are familiar with the topic which is understood by the name of the nodes. GPT-4o (FullDist) leads with the lowest median KL divergence, followed by Claude 3.5 Sonnet (FullDist) and Claude 3.5 Sonnet (SepState).

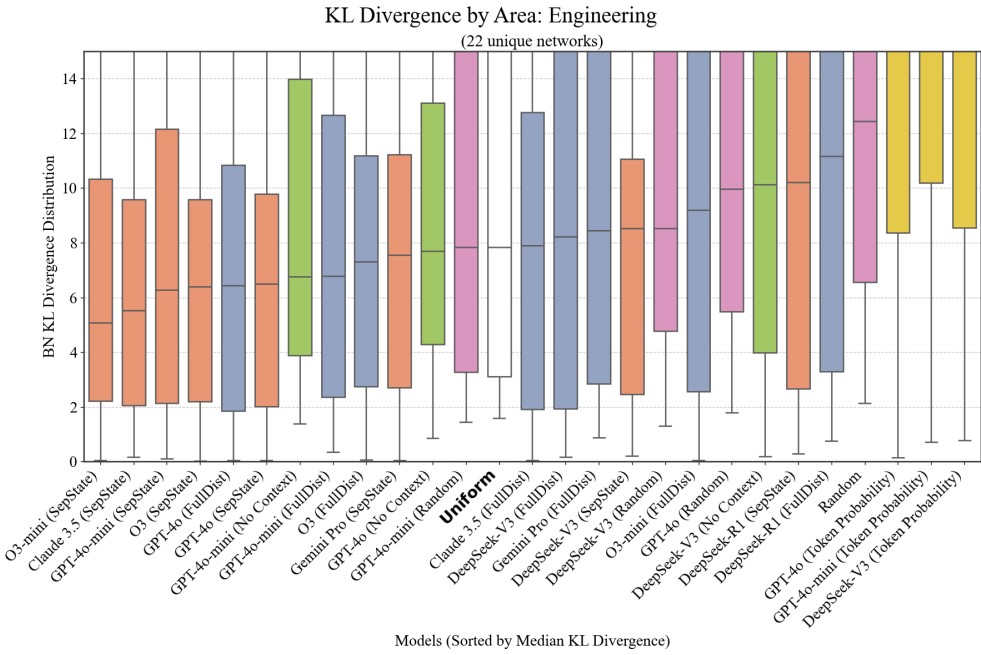

Figure 34: BN KL divergence by area: Engineering. As the largest domain in our dataset, this area offers the most reliable per-domain comparison. All o3 and GPT models perform really well in this domain with other models lagging behind the baseline.

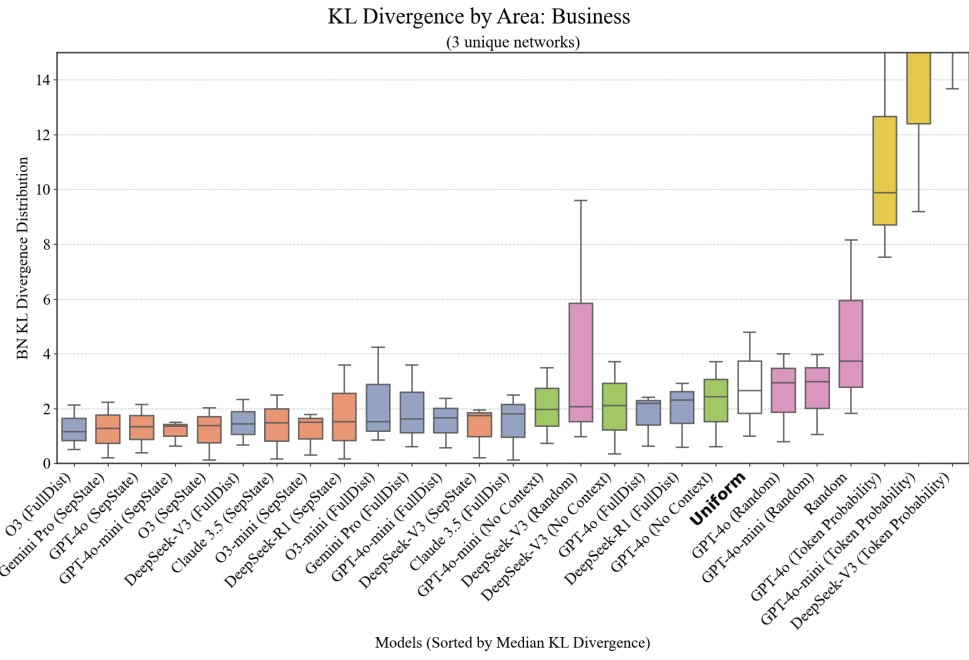

Figure 35: BN KL divergence by area: Business (3 unique networks). o3 (FullDist) and Gemini 1.5 Pro (SepState) achieve the lowest median KL divergence. However, with only 3 BNs, these results are suggestive.

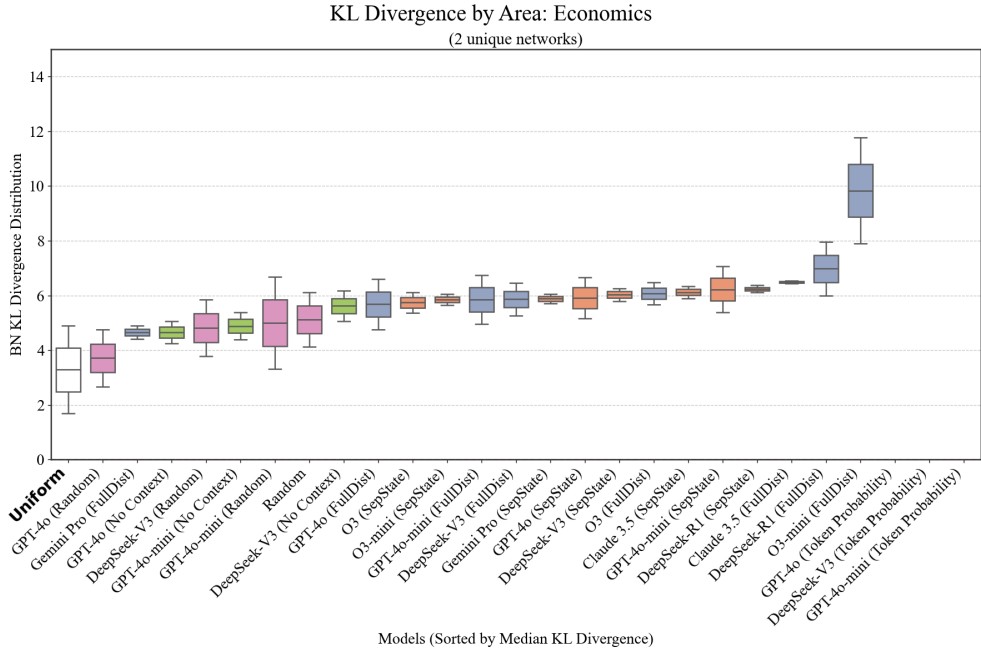

Figure 36: BN KL divergence by area: Economics. With only 2 BNs, no definitive conclusions can be drawn.

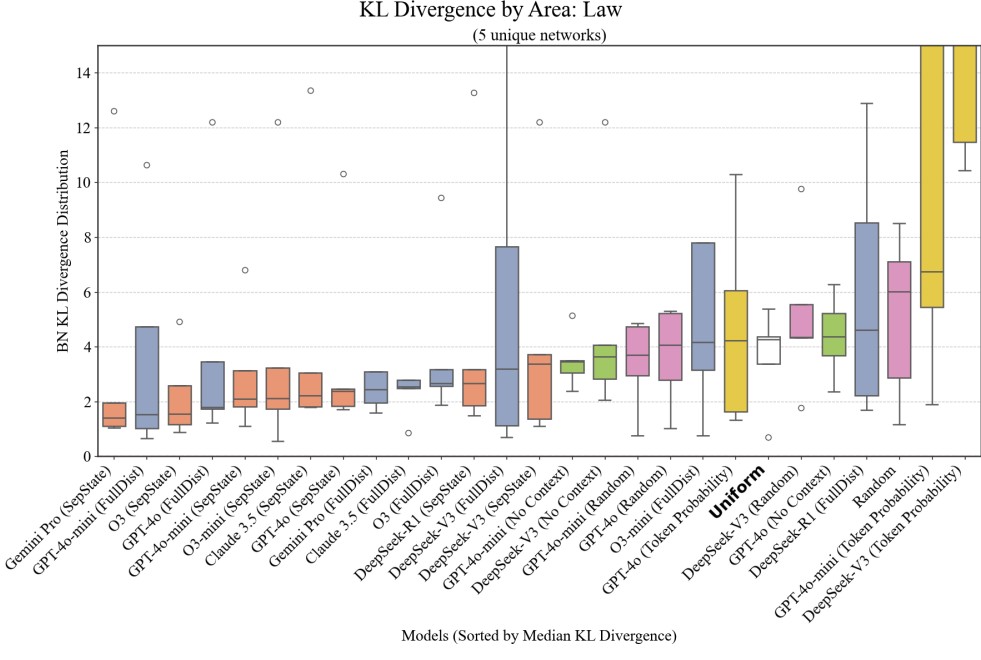

Figure 37: BN KL divergence by area: Law. Gemini 1.5 Pro (SepState) achieves the lowest median KL divergence, followed by GPT-4o-mini (FullDist) and o3 (SepState). With only 5 BNs, these findings are suggestive and represent an educated guess.

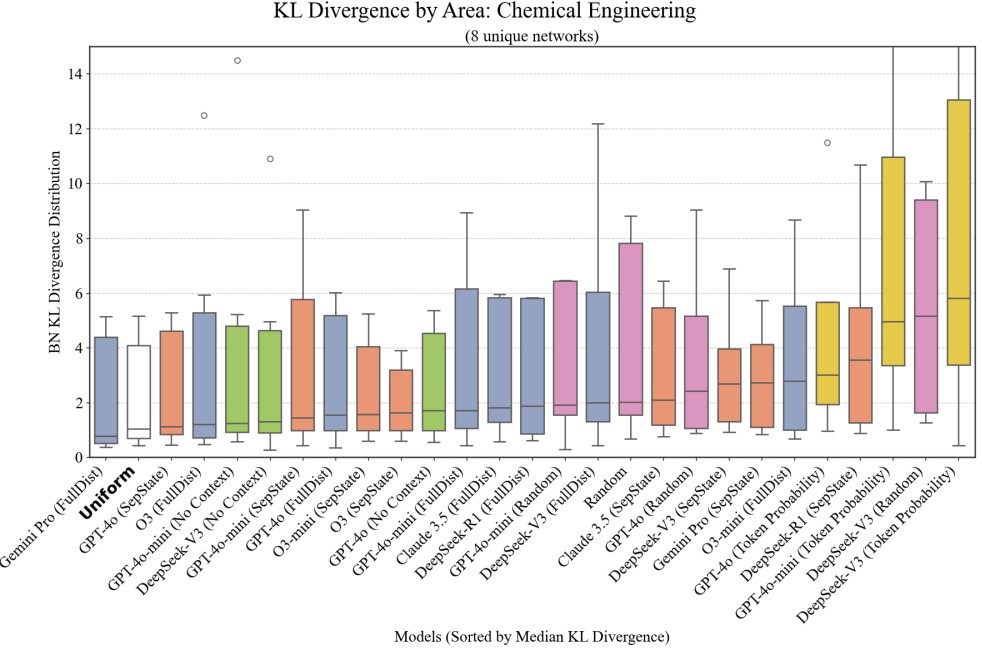

Figure 38: BN KL divergence by area: Chemical Engineering. Gemini 1.5 Pro (FullDist) achieves the lowest median KL divergence, with GPT-4o (SepState) also performing competitively but still worse than baseline.

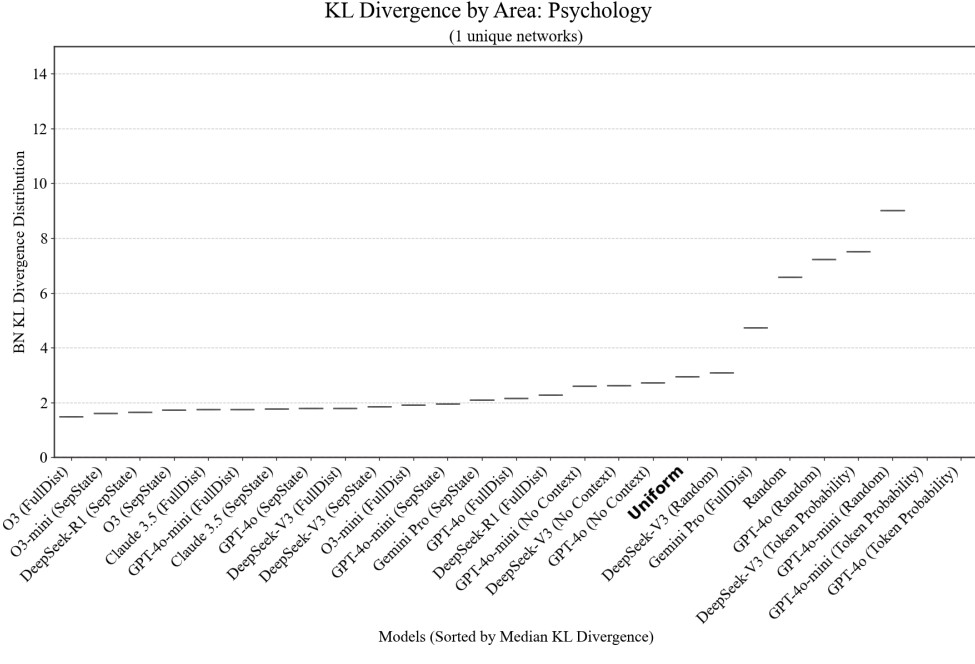

Figure 39: BN KL divergence by area: Psychology. With only 1 BN, no definitive conclusions can be drawn.

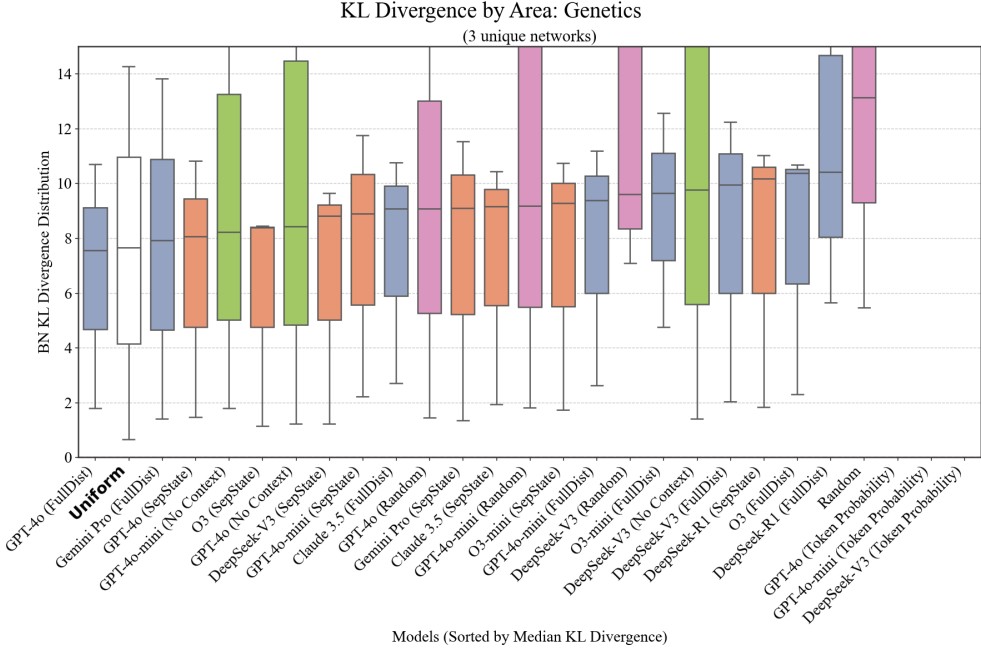

Figure 40: BN KL divergence by area: Genetics. With only 3 BNs, no definitive conclusions can be drawn.

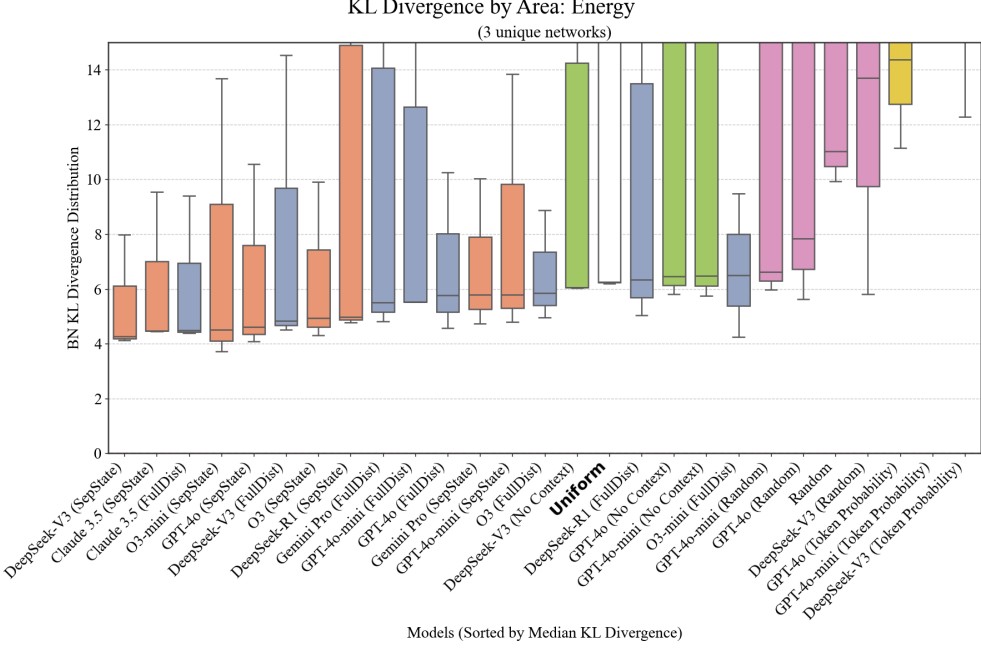

Figure 41: BN KL divergence by area: Energy. With only 3 BNs, no definitive conclusions can be drawn.

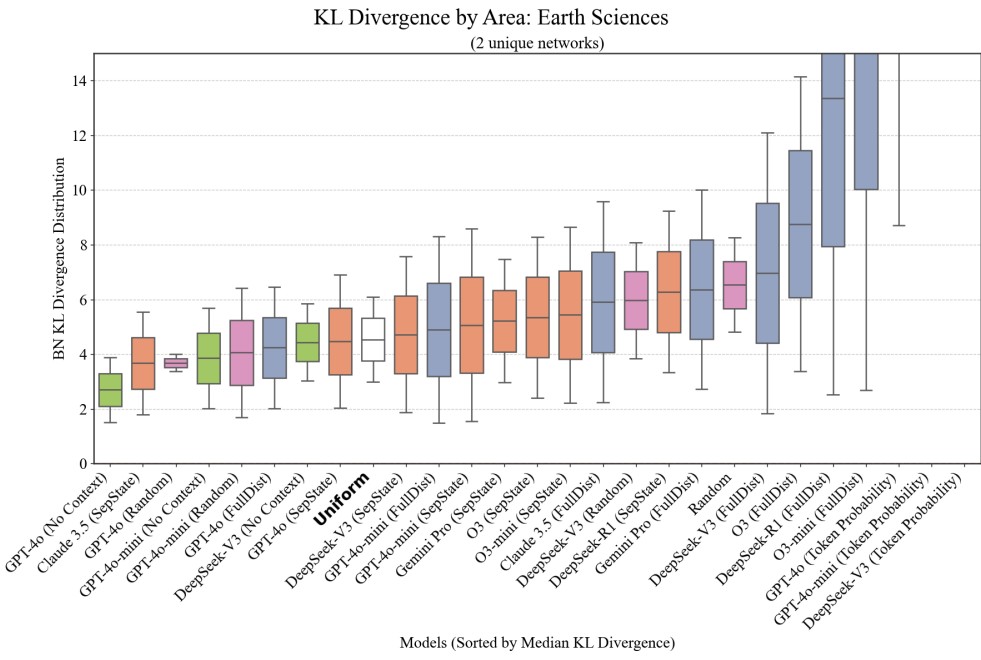

Figure 42: BN KL divergence by area: Earth Sciences. With only 2 BNs, no definitive conclusions can be drawn.

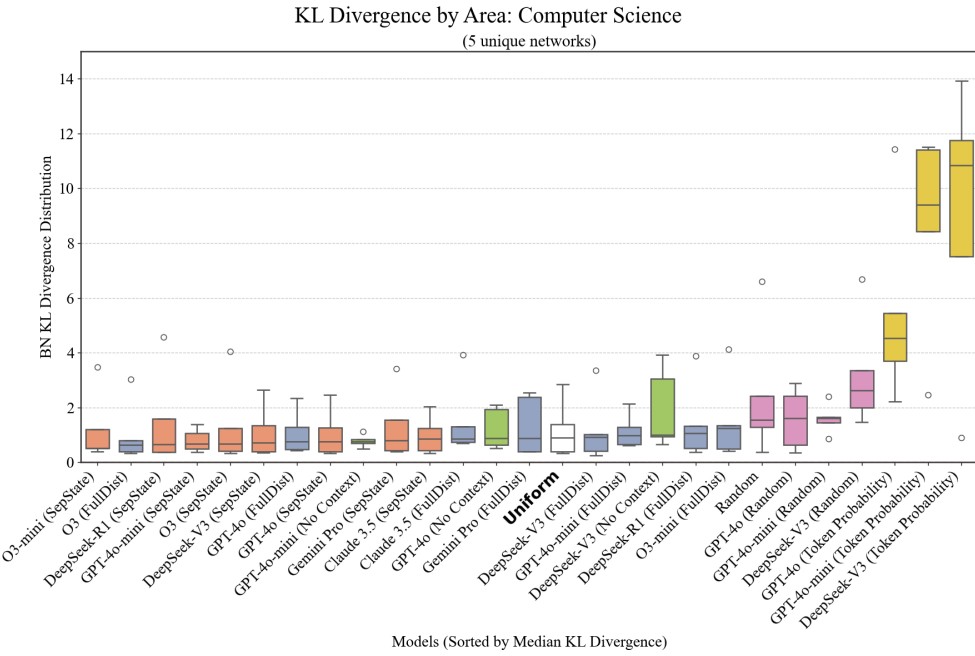

Figure 43: BN KL divergence by area: Computer Science. With 5 BNs, these findings are suggestive, though the consistently low KL divergence values indicate that LLMs perform well on computer science BNs.

