# OpenReview forum: "Extracting Probabilistic Knowledge from Large Language Models for Bayesian Network Parameterization"
_TMLR — Accepted by TMLR_

### Review · Reviewer_19WK · 2026-02-17

**Summary Of Contributions:**

The paper investigates the use of LLMs for prior parameterisation in Bayesian networks (BNs). For this, the authors assume the BN structure is given, and only the prior parameterisation needs to be predicted. The LLM is given instructions and context on the random variables in the BN. The resulting probabilities are then normalised and used as prior information in discrete-state BNs, assuming a conjugate prior (Dirichlet-multinomial), e.g., as pseudo-data counts. To generate the prior parameterisation, the authors test two approaches: (i) prediction of the prior probabilities for the join, (ii) prediction of each conditional given by the BN structure. The authors further evaluate the proposed method against uniform and randomly generated priors and maximum-likelihood estimation across varying numbers of observations. Methods are evaluated on 80 BNs selected from the bnRep repository. The proposed method tends to provide slight improvements over uniform priors in low-data regimes and performs equally well with more observations. Significant improvements are not observed in the box plots provided; however, the statistical test results are not provided. Lastly, the authors evaluate their approach on classification tasks and find that it performs comparably or better than maximum likelihood estimation.

**Main Strength:** The paper is topical and using LLMs as a surrogate for human expert knowledge is an intruiging research questions.

**Main Weakness:** The actual contribution is very limited and the results are not convincing. In fact, in most experiments, a uniform prior baseline seems to work as well as the LLM-generated prior specifications. Given that the use of LLMs entails substantial additional costs and yields marginal or no gains in this setting, it is questionable how useful the proposed method is. In addition, the paper overstates the gains of the proposed method several times.

**Audience:**

Yes

**Audience Explanation:**

The research is topical and a potentially interesting study for those working on Bayesian networks and probabilistic modelling.

**Broader Impact Concerns:**

As the method uses LLMs to extract (domain) knowledge for prior specification, and those LLMs might reflect biases against ethnicities or minority groups, the BN prior specification might, in consequence, incorporate those negative biases. The work currently does not reflect on its broader impact and ethical concerns, and a detailed discussion would be necessary.

**Claims And Evidence:**

No

**Claims Explanation:**

While the claims about the technical details of the present method are accurate, the conclusions drawn from the results are inaccurate. In particular, the authors repeatedly state that LLM-generated prior parameterisations surpass the baselines; however, I do not find that the results presented support this conclusion.

**Requested Changes:**

- **Vetting of LLMs:** The authors note that one challenge of using human experts is that their competence must be vetted. I was wondering what the authors think about using LLMs, would their competence also need to be vetted, and should this be done?
- **Prior Elicitation, missing related work:** The current related work exposition is missing several citations and a discussion of how the proposed work relates to those works, e.g., including but not limited to Li et al. (2023); Selby et al. (2025); Capstick et al. (2024).
- **Zero-shot Regression, missing related work:** Similarly, there seems to be missing related work on LLM-based regression, such as Requeima et al. (2024), which uses LLMs to perform probabilistic regression. The paper would benefit from a more complete discussion of related literature.
- **Page 5, minor comment:** It would be good to specify more precisely how $\alpha$ is defined. I assume it is a positive finite real number including zero, i.e., $\alpha \in [0,\infty)$.
- **Section 5.1:** How is it ensured that the ground through BN parameterisation is not included in the training data of the LLM? Given that this is an open-source dataset, information about it is likely included in the training corpus of recent frontier LLMs.
- **BN KL divergence:** The authors use what they call the BN KL divergence to compare estimated Bayesian networks with ground truth through parameterisations. From my understanding, when following the equations in the appendix, this is just the KL between BNs as presented by Koller & Friedman (2009), Section 8. Note that computing the KL (or what authors call the BN KL) is intractable in general for BNs even if they share the same structure, as it requires computation of the marginals $p(\operatorname{pa}_i)$, which is only tractable (efficient) in constrained settings, e.g., low-tree width BNs. Even though this is conceptually not an issue, _it requires a clear presentation that this decomposition does not solve the computational challenge of computing the KL divergence in BNs_. At the moment, the text reads as if the BN KL is some special way of computing the KL that avoids computational blow-up in BNs.
- **CPT KL divergence:** I believe the presentation of the CPT KL divergence is a bit confusing. From what I understand, this is the KL divergence of the individual conditional distributions defined by the BN structure, aggregated. This is totally fine, but it needs a clear presentation and discussion that this is only computable efficiently in certain cases, e.g., the number of rows can be exponentially large. It would also be helpful to provide an in-depth discussion of why this metric is meaningful.
- **Section 5.3:** I was expecting a more thorough discussion of why the FullDist approach underperforms and what other approaches could be valuable directions to explore.
- **Section 5.4:** The authors claim that EDP "consistently outperform[s] [...] Uniform", but I do not think this conclusion can be drawn from Figure 2. If anything, it seems clear that Uniform priors work comparably to those predicted by the LLM (e.g., EDP). Only in the low data regime (10 or 3 observations) can a difference be observed.
- **Uniformity:** The previous remark raises the question, how different are those prior probabilities predicted by the LLMs from uniform priors? Could the authors provide an ablation on how informative the priors actually are?
- **Results:** No statistical hypothesis test results are provided for both results presented in Section 5.3 and Section 5.4.
- **Table 1:** Results should only be bolded if they are statistically different from the second best (e.g., MLE).


### References

- Alexander Capstick, Rahul G Krishnan, and Payam Barnaghi. Autoelicit: Using large language models for expert prior elicitation in predictive modelling. arXiv preprint arXiv:2411.17284, 2024.
- Daphne Koller and Nir Friedman. Probabilistic graphical models: principles and techniques. MIT press, 2009.
- Belinda Z Li, Alex Tamkin, Noah Goodman, and Jacob Andreas. Eliciting human preferences with language models. arXiv preprint arXiv:2310.11589, 2023.
- James Requeima, John Bronskill, Dami Choi, Richard Turner, and David K Duvenaud. LLM processes: Numerical predictive distributions conditioned on natural language. Advances in Neural Information Processing Systems, 37:109609109671, 2024.
- David Selby, Yuichiro Iwashita, Kai Spriestersbach, Mohammad Saad, Dennis Bappert, Archana Warrier, Sumantrak Mukherjee, Koichi Kise, and Sebastian Vollmer. Had enough of experts? quantitative knowledge retrieval from large language models. Stat, 14(2):e70054, 2025.

---

> ### Author Response · Authors · 2026-03-21
>
> Thank you for your time, thorough review, and constructive feedback. We appreciate your insights, which have helped us improve the clarity and rigor of our paper. Below, we address your comments and outline the corresponding revisions made to the manuscript.
>
> **Main Weakness:** To address your primary concern regarding the marginal gains over a uniform prior and the cost-benefit tradeoff of using LLMs, we have conducted a statistical significance analysis on the results in Section 5.3. Because our evaluation spans 80 different Bayesian Networks, the aggregate results demonstrate a statistically significant advantage over the uniform baseline. We have added the following analysis to the paper:
> > A one-sided paired Wilcoxon signed-rank test, comparing each model's per-BN KL divergence against the uniform baseline, confirms statistical significance at $p<0.01$ for GPT-4o (SepState and FullDist), o3-mini (SepState), o3 (SepState), Claude 3.5 (SepState), and GPT-4o-mini (SepState). Gemini-Pro (SepState and FullDist) achieves $p<0.05$, while the remaining configurations that outperform uniform, namely GPT-4o-mini (FullDist) and DeepSeek-V3 (SepState and FullDist), achieve $p<0.1$.
>
> Regarding the substantial additional costs of LLMs, we have expanded our discussion section to discuss different prompting configurations and note that the reasoning steps of the LLMs can be omitted, which yields comparable results while significantly reducing inference costs. Additionally, the FullDist approach can be utilized as a cheaper alternative, while it slightly trails SepState in performance, it stil outperforms the uniform baseline.
>
> **Vetting of LLMs:** We agree that vetting LLM competence is an important consideration. We have added a new section titled `Limitations and Ethical Considerations`, which discusses this.
>
> **Missing Related Work:** Thank you for pointing out these relevant studies. We have incorporated and discussed Selby et al. (2025), Capstick et al. (2024), and Requeima et al. (2024) in the related work section. We reviewed Li et al. (2023), but because its primary focus is on eliciting human preferences rather than probabilistic priors, we found it only loosely related to our specific problem formulation and opted not to include it. If you feel we have misinterpreted its relevance, please let us know and we would be happy to add it.
>
> **Definition of $\alpha$:** The range of $\alpha$ is specified in Section 4.2. Furthermore, in Section 5.2, we direct readers to the Appendix for a deeper analysis and further details regarding the definition and behavior of $\alpha$.
>
> **Data Contamination:** We acknowledge that, as with many studies using LLMs, we cannot definitively rule out data contamination from the dataset. However, our core contribution remains independent of this contamination. Our focus is on how to best extract and utilize the information provided by the LLM, whether by using it as a prior, using it directly, or combining it with observed data. Even if the LLMs have perfectly memorized the ground truth, effectively integrating that information as a prior distribution is a non-trivial challenge that our method addresses.
>
> **BN and CPT KL Divergence Clarifications:** We have updated the Appendix to explicitly state that our BN KL decomposition does not solve the general computational blow-up of computing the KL divergence in BNs. However, in practice, computation was highly efficient for our experiments using pgmpy. Regarding the CPT KL divergence, we have clarified in the Appendix that while the number of rows can be exponentially large in theory, the realistic BNs utilized in our study are designed with efficient structures that mitigate this issue.
>
> **EDP vs. Uniform Baseline:** As demonstrated by the newly added statistical tests in Section 5.3, the LLM-generated priors do consistently outperform the uniform priors overall. In Figure 2 (Section 5.4), EDP shows a clear advantage in the low-data regimes. As the number of samples increases, the value of any prior naturally decreases, and it is expected behavior that the performance gap between EDP and Uniform narrows as data grows, ultimately rendering the choice of prior negligible given enough data.
>
> **Table 1:** Because we calculate statistical significance for the entirety of the results as an aggregate (rather than row-by-row for individual networks), we cannot confidently bold individual cells to denote statistical significance against the second-best method.
>
> Thank you once again for your thorough review and constructive comments. We hope these revisions and clarifications address your concerns and improve the overall quality of the manuscript.

---

> ### Comment · Reviewer_19WK · 2026-03-21
>
> Thank you for the response.
>
> > we have conducted a statistical significance analysis
>
> As you are using a frequentist approach here (I believe the authors might want to use a paired test rather than a one-sided test), I would also suggest reporting the Bayes factor instead.
>
> Would it be possible to report the results here for better visibility?

---

> > ### Author Response · Authors · 2026-03-22
> >
> > Thank you for the suggestions. Below are the full results of the one-sided and two-sided  Wilcoxon signed-rank test. We also included the Bayes factor (BF10) as recommended.
> >
> >
> > | # | Model | p (one-sided) | p (two-sided) | BF10 |
> > |---|-------|---------------|---------------|-------|
> > | 1 | GPT-4o (SepState) | **0.0002 (<0.01)** | **0.0003 (<0.01)** | **10.44 (>3)** |
> > | 2 | GPT-4o-mini (SepState) | **0.0045 (<0.01)** | **0.0091 (<0.01)** | **9.32 (>3)** |
> > | 3 | GPT-4o (FullDist) | **0.0002 (<0.01)** | **0.0004 (<0.01)** | **5.21 (>3)** |
> > | 4 | O3-mini (SepState) | **0.0012 (<0.01)** | **0.0025 (<0.01)** | **4.27 (>3)** |
> > | 5 | Claude 3.5 (SepState) | **0.0037 (<0.01)** | **0.0073 (<0.01)** | **3.28 (>3)** |
> > | 6 | O3 (SepState) | **0.0027 (<0.01)** | **0.0055 (<0.01)** | **6.96 (>3)** |
> > | 7 | GPT-4o-mini (FullDist) | 0.0630 (<0.1) | 0.1260 | 1.39 (>1) |
> > | 8 | DeepSeek-V3 (FullDist) | 0.0896 (<0.1) | 0.1793 | 0.15 |
> > | 9 | Claude 3.5 (FullDist) | 0.1468 | 0.2935 | 0.13 |
> > | 10 | O3 (FullDist) | 0.1614 | 0.3228 | 0.36 |
> > | 11 | DeepSeek-V3 (SepState) | 0.0814 (<0.1) | 0.1628 | 0.58 |
> > | 12 | Gemini Pro (SepState) | **0.0304 (<0.05)** | 0.0607 (<0.1) | 0.78 |
> > | 13 | Gemini Pro (FullDist) | **0.0109 (<0.05)** | **0.0218 (<0.05)** | 0.13 |
> >
> > The table is sorted by median KL divergence, as in the figure in the paper.
> >
> > The two-sided paired Wilcoxon signed-rank test confirms statistical significance at $p<0.01$ for GPT-4o (SepState and FullDist), o3-mini (SepState), o3 (SepState), Claude 3.5 (SepState), and GPT-4o-mini (SepState). Gemini Pro (FullDist) achieves $p<0.05$ under the two-sided test, while Gemini Pro (SepState) achieves $p<0.1$.
> >
> > With the Bayes factors GPT-4o (SepState), GPT-4o-mini (SepState), o3 (SepState), GPT-4o (FullDist), o3-mini (SepState), and Claude 3.5 (SepState) all achieve BF10$ > 3$, showing moderate-to-strong evidence in favour of the alternative hypothesis.
> >
> > The top-performing models outperform the uniform baseline using either statistical significance tests.

---

> > > ### Comment · Reviewer_19WK · 2026-04-07
> > >
> > > Thank you very much for the additional results!
> > >
> > > A part of my main concern has hereby been addressed; it remains the question about the utility of this approach, as it introduces substantial additional costs by using LLMs while providing moderate improvements and adds an additional layer that can produce unreliable/hallucinated results, biasing the final results.

---

> > > > ### Author Response · Authors · 2026-04-13
> > > >
> > > > Thank you for the follow-up and for acknowledging the additional results.
> > > >
> > > > We would like to add a further perspective regarding the cost of utilizing LLMs. While our previous response highlighted methods to reduce these costs, it is equally important to consider the context-dependent value of the performance gains. In certain applications, even marginal improvements can easily justify a higher financial or computational expense. Crucially, the cost of LLMs should not be evaluated only against a uniform prior, but rather against the real-world alternatives: 1) conducting large-scale data collection or 2) eliciting knowledge from domain experts. Gathering sufficient domain-specific data is not only expensive but often infeasible at scale. Domain experts are also expensive and not always available.
> > > >
> > > > Furthermore, while we acknowledge the risk of LLM hallucinations, our approach still demonstrated consistent improvements across most of our evaluated BNs. It is also worth noting that expert elicitation is not a flawless alternative. Assuming a domain expert is both available and cost-effective, there is no guarantee they would definitely outperform the model. As discussed in our paper, relying on human experts introduces its own set of practical challenges, such as the vetting required and the complexities of reconciling conflicting views among multiple experts.
> > > >
> > > > Therefore, despite the inherent noise and costs associated with LLMs, their ability to consistently improve upon baseline priors makes them a highly valuable tool in settings where additional data and experts are even more expensive or not available.

---

### Review · Reviewer_Zux9 · 2026-02-25

**Summary Of Contributions:**

This paper investigates whether large language models (LLMs) can serve as sources of probabilistic expert knowledge for parameterizing Bayesian Networks (BNs). It introduces a large-scale evaluation framework for extracting conditional probability tables (CPTs) from LLMs and proposes two prompting strategies for eliciting full conditional distributions. The experimental analysis supports the main claims of the paper.

Strengths:
- The evaluation across 80 different real-world BNs is substantially broader than prior work, which often relies on toy networks or shallow binary settings.
- The use of BN KL divergence enables comparison with ground-truth CPTs without relying solely on accuracy.
-  A good ablation study.

Weaknesses
- More details are provided in the appendices than in the main paper. Two of the three strengths are explained primarily in the appendices.
- Heavy reliance on prompt engineering; robustness across prompting variations is underexplored.

**Additional Comments:**

Section 3 is too short and provides too little information for the paper.

**Audience:**

Yes

**Audience Explanation:**

The paper addresses an interesting problem, presenting a substantial experimental campaign and a good discussion of the results in a field where few works consider the same task.

**Broader Impact Concerns:**

The work proposes using LLMs as probabilistic experts in domains including healthcare, engineering, and finance.

Potential concerns include:
- Results may be affected by hallucinations or biases and could be risky without proper validation, particularly in sensitive domains such as healthcare.
- The reasoning processes of LLMs cannot be audited, making it difficult to ensure the correctness and reliability of the estimated values.

**Claims And Evidence:**

Yes

**Claims Explanation:**

The central claims are convincingly supported. However, there are some drawbacks for some of them.
For example, the paper lacks a statistical significance test for KL divergence comparisons across models making less robust the claim about the ability of LLMs to estimate correct probabilities.

**Requested Changes:**

A more detailed discussion of the role of \alpha should be included in the paper, along with a sensitivity analysis over its values.

I would also suggest clarifying how normalization handles pathological outputs (e.g., negative numbers or out-of-range probabilities). Moreover, values not summing to one do not guarantee that the probabilities are provided on the same “scale” or under the same background assumptions. For example, if we obtain two probabilities equal to 0.9 and one equal to 0.4, can we be sure that the LLM is considering the same conditions for all of them? It could happen that the value 0.4 is actually correct, but normalization reduces it to enforce a proper probability mass. Using an example from the paper, what if, for one prompt, the LLM returns a probability of 0.9, while for another prompt asking about the probability that a person who smokes cigarettes will develop cancer in their lifetime it returns 0.2? The second value is correct following the literature but it will be normalized. Have I misunderstood some aspects of the method?

As already noted, there is a strong imbalance between what is reported in the main paper and what is presented in the appendix. The description of the preprocessing steps and the dataset is relegated to the appendix, and many results supporting key claims are also reported there. In my view, this distribution is overly unbalanced.

---

> ### Author Response · Authors · 2026-03-21
>
> Thank you for reviewing our paper and for your valuable suggestions.
>
> We agree that the role of $\alpha$ warrants more discussion. We conducted a sensitivity analysis over its values, now included in Appendix C.3, showing that our results are robust across different values of $\alpha$. We also added a reference to this in Section 5.2:
>
> > Refer to Appendix C for LLM and hyperparameter configurations, specifically detailing our rationale for selecting $\alpha$ and its impact on performance.
>
> We also clarified in Section 5.2 that our pipeline handles pathological or invalid outputs (e.g., negative numbers or out-of-range probabilities) by simply reprompting the LLM until a valid format is returned.
>
> You are correct that normalization can alter an individually "correct" isolated probability to enforce a proper probability mass across the distribution. However, our empirical results demonstrate that applying normalization works better in our experiments overall. While it might adjust specific true values in isolated prompts, in aggregate, it effectively calibrates the model's outputs and improves performance.
>
> To better balance the paper, we moved the dataset descriptions from the Appendix into the main text.
>
> We also share your broader impact concerns regarding the use of LLMs as probabilistic experts. To properly address these risks, we added a `Limitations and Ethical Considerations` section to the paper, which mentions the need for LLMs to be vetted for each domain and their limits.
>
> Finally, Section 3 is now expanded to explain things in more detail, with an example, and to discuss the classification task as well.

---

### Review · Reviewer_oy2G · 2026-03-06

**Summary Of Contributions:**

This paper evaluates the capability of LLMs to parameterize Bayesian Networks (BNs) by extracting probabilistic knowledge from their internal representations. The authors propose two prompting schemes (SepState and FullDist) to elicit conditional probability distributions from LLMs, and an Expert-Driven Priors (EDP) framework that combines LLM-derived priors with empirical data. Experiments on 80 real-world BNs show that LLM estimates outperform baselines (random, uniform, token probability), and EDP improves parameter estimation especially in low-data regimes.

**Audience:**

Yes

**Audience Explanation:**

The paper presents a novel application of LLMs for Bayesian Network parameterization, which would be of interest to researchers exploring unconventional uses of LLMs beyond text generation. The finding that LLMs can serve as reasonable expert priors, and the counterintuitive result that reasoning models perform worse, would appeal to a broad audience.

**Broader Impact Concerns:**

LLMs are prone to hallucination, and using their outputs as probabilistic priors in critical domains (e.g., healthcare, safety engineering) could lead to incorrect probability estimates that propagate through downstream decision-making. The paper does not sufficiently discuss this risk or propose safeguards for deployment in high-stakes settings.

**Claims And Evidence:**

Yes

**Claims Explanation:**

The experimental evaluation is extensive and well-designed, covering 80 real-world BNs, multiple LLMs, diverse baselines, and both intrinsic (KL divergence) and extrinsic (downstream classification) metrics. The claims are generally well-supported by the evidence presented.

**Requested Changes:**

- The paper does not sufficiently analyze the reliability of LLM internal knowledge across different types of domains. For commonsense-level relationships (e.g., smoking and lung cancer), LLMs are likely to produce reasonable estimates. However, for specialized domains where pre-training data may be sparse or contested (e.g., the relationship between COVID vaccination and cardiac events) the LLM's internal knowledge may be unreliable or even harmful as a prior. The paper lacks a breakdown of the 80 BNs into commonsense vs. specialized categories and does not analyze whether LLM priors underperform uniform baselines in specific domains. More importantly, there is no discussion of augmenting LLMs with external reliable sources (e.g., web search, PubMed) to ground their estimates in evidence. Such a RAG-based pipeline would be essential for deploying this approach in specialized domains, and the absence of this discussion is a notable gap.

- The paper does not include a sensitivity analysis on the specific prompt wording used in Figure 3. Since LLMs are known to be sensitive to prompt phrasing, it is unclear how robust the probability estimates are to variations in the prompt template (e.g., alternative phrasings, different instructions for reasoning, etc.).

---

> ### Author Response · Authors · 2026-03-21
>
> Thank you for reviewing our paper and your valuable suggestions.
>
> We have added a new appendix section (Appendix F: "BN KL Divergence by Domain Area") that breaks down performance across all 13 domain areas covered by our 80 BNs: Social Sciences (11 BNs), Environmental Science (8 BNs), Medicine (7 BNs), Engineering (22 BNs), Business (3 BNs), Economics (2 BNs), Law (5 BNs), Chemical Engineering (8 BNs), Psychology (1 BN), Genetics (3 BNs), Energy (3 BNs), Earth Sciences (2 BNs), and Computer Science (5 BNs). This section notes the top-performing model(s) and sample size of each area. This analysis reveals several noteworthy patterns. In domains where LLMs are likely to have substantial training data, such as Medicine and Computer Science, most models comfortably outperform the Uniform baseline. In contrast, more specialized domains like Economics show the Uniform baseline performing competitively. These findings are consistent with the reviewer's intuition that LLM reliability varies with domain specialization. The Discussion section, `Performance Variations Among LLMs`, which previously hinted at this difference, now references this detailed appendix.
>
> Regarding the external-sources search, we leave the online exploration to future work. We initially planned to add showcases of this for a few BNs, but searching the web to find instances of nodes introduces a lot of complications. For example, the LLM might find the exact answer directly rather than just the examples. As a result, it is difficult to properly test and would require significant effort.
>
> Regarding prompt sensitivity, we have added a new discussion section, `Effect of Prompt Instructions on Probabilities`, which elaborates on this. Specifically, we tested several alternative prompt configurations, including instructing the LLM to provide only numerical answers without reasoning, and augmenting prompts with additional BN-level context beyond node descriptions. None of these variations affected the aggregate results across the 80 BNs. This robustness is consistent with related work on LLM-based regression, which similarly finds that chain-of-thought reasoning does not significantly alter predicted numerical outputs when aggregated over a dataset. The only factor that mattered was the definition of the nodes that we had already analyzed in our results.
>
> Finally, we have added a `Limitations and Ethical Considerations` section which directly addresses your concern. We argue that (1) LLM performance varies by domain, so the selected LLM must be vetted for the target domain before deployment; (2) outperforming a uniform baseline does not establish readiness for safety-critical applications, and in high-stakes domains like healthcare or industrial safety engineering, LLM-derived priors should be treated as strong initial estimates rather than definitive conclusions.
>
> We hope these additions address your concerns.

---

### Comment · Editors_In_Chief · 2026-05-31

On 5/30, by request of the authors, the EiCs replaced the camera ready version with a new revision. The authors report that the only difference is the addition of an acknowledgments section.

---

### Decision · Action_Editor_Jwim · 2026-04-30

**Recommendation:** Accept with minor revision

**Additional Comments:**

While all the reviewers lean towards acceptance of the submission as it fulfills the main acceptance criteria of TMLR, the reviews and discussions highlighted a few aspects that should be addressed in the camera ready version of the paper:

* Update the limitations sections in line with the comments and discussion with the reviewers
* Briefly discuss costs and benefits of the proposed approach over other approaches for BN parameterization at a central place in the paper (this is currently spread out in the submission but central for the motivation as highlighted in the author-reviewer discussion).

**Audience:**

Yes

**Audience Explanation:**

Bayesian networks are important probabilistic graphical models used widely within the community. However, their accurate specification can the be time consuming and expensive and approaches for making their specification simpler are thus interesting to parts of TMLR's audience.

**Claims And Evidence:**

Yes

**Claims Explanation:**

The paper was reviewed by three expert reviewers which all came to the conclusion that claims made in the paper are well supported by the presented evidence. In particular, the authors demonstrate on eighty Bayesian networks how well conditional probabilities of events can be specified through LLMs.